# Revealing the unseen: Benchmarking video action recognition under occlusion

**Shresth Grover**
IIT Kanpur
shrgo@iitk.ac.in

**Vibhav Vineet**
Microsoft Research
vibhav.vineet@microsoft.com

**Yogesh S Rawat**
University of Central Florida
yogesh@crcv.ucf.edu

## Abstract

In this work, we study the effect of occlusion on video action recognition. To facilitate this study, we propose three benchmark datasets and experiment with seven different video action recognition models. These datasets include two synthetic benchmarks, UCF-101-O and K-400-O, which enabled understanding the effects of fundamental properties of occlusion via controlled experiments. We also propose a real-world occlusion dataset, UCF-19-Y-OCC, which helps in further validating the findings of this study. We find several interesting insights such as 1) *transformers are more robust than CNN counterparts*, 2) *pretraining make models robust against occlusions*, and 3) *augmentation helps, but does not generalize well to real-world occlusions.* In addition, we propose a simple transformer based compositional model, termed as CTx-Net, which generalizes well under this distribution shift. We observe that CTx-Net outperforms models which are trained using occlusions as augmentation, performing significantly better under natural occlusions. We believe this benchmark will open up interesting future research in robust video action recognition. Code is publicly available at https://shroglck.github.io/rev_unseen.

## 1 Introduction

Video action recognition is a challenging and important task in computer vision with numerous real-world applications, including security, robotics, sports analysis, and human-computer interaction [42, 39]. Recent years have witnessed significant progress in this field, facilitated by the availability of large-scale datasets [20, 8, 40], enabling the learning of increasingly complex models [1, 7, 11, 13, 44, 30], which have demonstrated human-level performance. However, the robustness of such models for real-world applications, particularly in the presence of occlusion, has not been explored [38, 6]. Given the dynamic nature of the environment, occlusion is an inherent property of real-world videos and needs to be studied.

To study this problem, we propose three benchmark datasets for analyzing the robustness of deep learning models against occlusion, namely UCF-101-O, K-400-O, and UCF-19-Y-OCC. The first two are derived from UCF-101(license CCLA-BY 4.0) [40] and Kinetics-400 (license CCLA-BY 4.0)[20, 8] respectively, and are created by synthetically occluding the actions in the videos using objects from the Pascal VOC dataset [10]. These benchmarks facilitate systematic analysis of occlusion effects by examining occluder properties such as severity and occluder dynamics. Additionally, we introduce a benchmark dataset with natural occlusion, UCF-19-Y-OCC, which comprises real-world videos containing natural occlusion. This dataset, is collected from YouTube, with categories based on a subset of classes from UCF-101, enables validation of observations made in the synthetic setting within a real-world environment.

We investigate the effects of occlusion on seven different video action recognition models, encompassing various aspects such as network size and architecture (transformer vs CNN). Through extensive

37th Conference on Neural Information Processing Systems (NeurIPS 2023) Track on Datasets and Benchmarks.

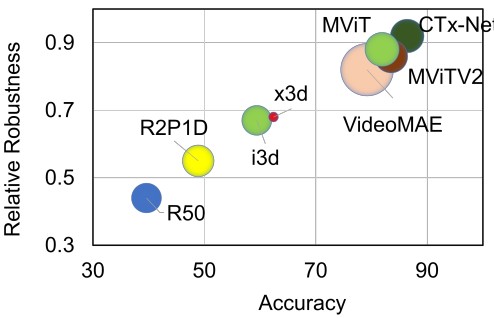 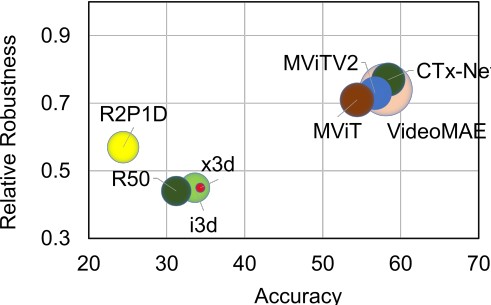

Figure 1: Comparison of relative robustness, accuracy, and model parameters for video action recognition models under occlusion. Left: UCF-101-O and right: K-400-O dataset. Size of circles indicate number of parameters in the model.

experimentation, we aim to assess the generalization capabilities of these models when confronted with both synthetic and natural occlusion. Our findings yield several interesting insights. Firstly, we observe that transformer based models exhibit significantly higher robustness to occlusion compared to CNN based models (Figure 1). Additionally, all models demonstrate superior resilience to temporally consistent forms of motion compared to more chaotic motion patterns. Natural occlusion likewise has a more detrimental impact on the robustness of CNN based models than transformer based models. Similarly, synthetic occlusion appears to disproportionately affect the performance of CNN-based models compared to transformer-based models. These experiments collectively illustrate a lack of generalization among current state-of-the-art models when faced with occlusion.

Data augmentation is a widely employed technique to address the issue of distribution shift. We investigate its impact on robustness of models against occlusion. Our experiments reveal a lack of generalization in models trained with data augmentations. Furthermore, we observe that such augmentations are insufficient in achieving generalization to real-world occlusion. We also propose CTx-Net, a compositional model for robust video action recognition under occlusion. Compositionality, helps captures interdependencies among parts, mimicking the robustness of the human visual system against distribution shifts [2, 15, 4]. Compositionality has shown good robustness properties in image-based tasks such as classification and object detection [24, 25, 22, 23, 41, 5], and we extend this to videos with the help of transformers. To validate its effectiveness, we extensively evaluate it on the proposed benchmark datasets, demonstrating its robustness in video action recognition under occlusion across both synthetic and natural scenarios.

We make the following contributions in this work,

- We study the problem of video action recognition under occlusion; this is the first work focusing on this problem to the best of our knowledge.
- We propose three benchmark datasets, K-400-O UCF-101-O, and UCF-19-Y-OCC, to study occlusion in video action recognition. K-400-O and UCF-101-O are created with the help of synthetic occlusions for a systematic study and UCF-19-Y-OCC consists of videos with real-world occlusions.
- We develop a simple transformer-based compositional model CTx-Net which generalizes well under occlusion.

## 2   Related work

**Video action recognition**   Current state-of-the-art video action recognition can be essentially broken down into two distinct ways. Firstly, CNN based models,[45, 49, 13, 18, 34, 12].Secondly, models that are based on the vision transformer[1, 31, 11, 28]. The majority of the currently available CNN-based methods make use of 3D CNNs. However, 3-D filters typically have many parameters and require the use of large-scale datasets such as Kinetics [20] in order to be trained effectively. As a result of the restricted receptive field of CNN architectures in general, they are unable to accurately model motion, which causes their performance to deteriorate as a direct result of this limitation.

In recent years, transformer architecture [46] has steadily gaining popularity as a direct result of the successes of vision transformers. Since then, there has been a surge in the amount of work done in this field, with models such as MViT, MViTv2, Swin [11, 28][31] and Timesformer[3] being able to give state-of-the-art performance on video-related tasks without significantly increasing the amount of memory overhead. [33] demonstrates the robustness of vision transformers on related tasks in the image domain, which inspired us to use transformer-based models in the video domain. Recently, [44] showed the ability of transformer models to run on relatively fewer data.Recently proposed [61, 17, 56, 57, 60]show that data augmentation is helpful in increasing robustness and data efficiency of video action recognition models.

**Occlusion reasoning** The existing works in image domain focus on occlusion for classification and image segmentation. In [14], the authors proposed using binary variables to infer visible cells. Hsiao and Herbert[19] modelled occlusion by using 3D relationship of objects with corresponding bounding boxes. Recent works on pixel level occlusion model use probabilistic methods as proposed by George et al. [16]. Another probabilistic approach by Yang et al.[54] introduces occlusion prior modeled by Markov random field in the domain of object tracking. Tighe et al. [43] introduce an inter class occlusion prior to parse scenes and refine pixel level labels. OFNET by Lu et al. [32] considers relevance between occlusion contours and pixel orientation Kortelewski et al. [22, 23] proposed a deep convolution based compositional network for occlusion reasoning at high level features. [58] studies the problem of scene de occlusion by obtaining the ordering graph of objects in the scene.In the context of the video domain, the challenge of occlusion has been investigated within the realms of pose estimation and video instance segmentation. Addressing this challenge, Li et al. [27] delve into the realm of action recognition under occlusion by leveraging radio signals to estimate poses. Meanwhile, Cheng et al. [9] tackle the issue of occlusion-induced ambiguity in 3D pose estimation by employing optical flow to evaluate the status of keypoints. Yang et al. [53] adopt a self-supervised paradigm to enhance pose estimation. Qi et al. [35] introduce a dedicated dataset specifically tailored for occluded instance segmentation. Ke et al.[21] introduce BCNet, which introduces a novel branch for inferring occluder information. In a similar vein, Lazarow et al. [26] introduce OCFusion, introducing an occlusion-aware module to signify occlusion relationships among mask proposals. Zhang et al.[60] propose a self-supervised approach that aims to estimate occlusion by recovering the temporal ordering of objects within videos.

Recently, there have been various efforts to combine compositional models with deep neural networks. Liao et al. [29] proposed to integrate compositionality into DCNNs by regularizing feature representations of DCNN's to cluster during learning. Zhang et al. [59] demonstrated that part detectors emerge in DCNNS by restricting the activations in feature maps to have localized distributions. However, these approaches have not been shown to enhance the robustness of deep models to occlusion. In [24, 22], the authors propose a CNN based compositional model for robustness against occlusion in images.

# 3 Benchmarking occlusion

In order to assess the robustness of current state-of-the-art models for video action recognition, we introduce three benchmarks. These datasets encompass a range of occlusion scenarios, including both synthetic and natural occlusions. The first two benchmarks, UCF-101-O and K-400-O, are synthetically curated and enable controlled studies of occlusion and serve as extreme cases for evaluating the impact of occlusion on action recognition models. Additionally, we introduce UCF-19-Y-OCC, a dataset comprising real-world videos with natural occlusions. This dataset provides an opportunity to examine the performance of models under more realistic occlusion conditions.

## 3.1 Design parameters

We study three key properties of occlusion: the type of occluder (including its shape and object class), the severity of occlusion (measuring the extent to which each frame is occluded), and the dynamics of the occluder (capturing its motion characteristics). By analyzing these properties, we aim to gain a comprehensive understanding of the impact of occlusion on video action recognition.

**Type of occluders** To ensure diversity among the occluders, we select 50 random objects from the PASCAL VOC dataset [10]. The objects are then masked out using segmentation masks, resized to

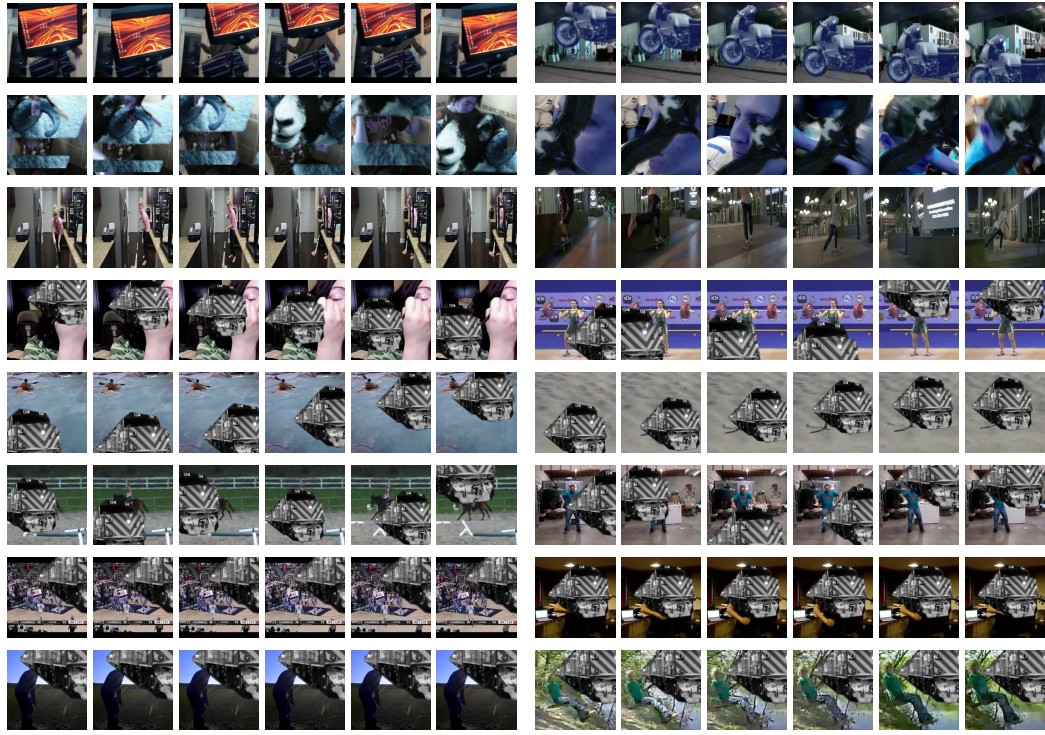

Figure 2: Samples video frames from proposed occluded benchmark datasets. First row: UCF101-O (left: '*Drumming*' and right: '*Jump Rope*') , second row: K-400-O (left: '*Air Drumming'* and right: '*Arm Wrestling*') , and third row UCF-19-Y-OCC (left: '*Mopping*' and right: '*Skate Boarding*'). Rows: 4,5,6,7,8-: Represent occlusion variations. Row 4: Represents occluder exhibiting linear motion, Row 5: Represents occluder exhibiting circular motion , Row 6 : Represents occluder exhibiting random motion , Row 7: Represents static occluder with severity 20-40% , Row 8 : Represents static occluder with 40-60% occluder severity

the desired dimensions, and finally pasted onto the frame at specific locations. This process allows us to maintain variability and realism of occlusion in environment.

**Severity of occlusions** In our study, we consider a range of occlusion severities from 0-60%. The severity of occlusion is quantified by calculating the average number of pixels occupied by the occluder throughout the video. For the proposed benchmark datasets, occlusion severities of 40-60% are employed, while severities of 0-40% are used during the training phase for augmentation experiments. This approach allows us to assess the impact of varying occlusion levels on video action recognition performance.

**Dynamics of occluders** We examine both static and dynamic occluders. For static occluders, the position of occluder is randomly chosen and remains constant throughout the video. To simulate dynamic motion, we employ four motion types with varying temporal coherence: linear, circular, and random. For linear motion, the occluder starts at a randomly selected coordinate and follows a linear path with a randomly determined slope. In the case of random motion, the occluder's position is randomly selected without considering its previous frame position. For circular motion, the occluder follows a circular path centered around a randomly chosen point. The center point is selected randomly for each video and remains consistent across all frames. In the benchmark datasets, we utilize static, random, and circular motions, while linear motion is used for augmentations. This allows us to assess the impact of different occluder dynamics on action recognition performance.

## 3.2   Benchmark datasets

We introduce three benchmark datasets to comprehensively investigate the challenge of occlusion in video action recognition. **UCF-101-O** consists of 3783 videos distributed across 101 action classes.

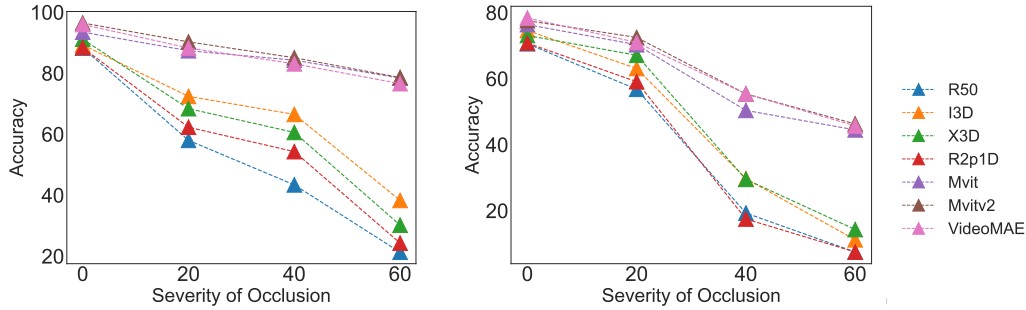

Figure 3: Variation of performance of action recognition models with changing occlusion severity on the proposed benchmark datasets. Left: UCF-101-O, and right: K-400-O dataset.

This includes all the videos in the test split 1 of UCF-101. The dataset incorporates occlusions that vary in size between 40-60%. The motion of occluders is randomly chosen from three categories: random motion, circular motion, and static. The occluders are selected randomly from a pool of 50 objects sourced from PASCAL VOC [10]. **K-400-O** consists of 40,000 video clips distributed across 400 action classes. It also includes all the testing videos from Kinetics-400. The selection of occlusion parameters and properties are same as UCF-101-O.

**UCF-19-Y-OCC** focuses on real-world occlusions and consists of publicly available videos from YouTube that feature natural occlusion. The dataset includes a total of 19 classes selected from the UCF-101 dataset, with each class consisting of 30 video clips of 5 seconds duration. The selection of classes was based on the likelihood of natural occlusion occurrence, while ensuring a diverse representation of actions to avoid bias towards specific classes.We conducted targeted searches on YouTube to curate a collection of videos pertinent to the specified class. From these we kept only those instances which contained occluded action sequences. We then extracted 5-second clips, within this duration, clips were incorporated into our dataset only if they encompassed a minimum of 2 seconds during which the action remained occluded. Figure 2 show sample video frames from the UCF-101-O, K-400-O dataset and UCF-19-Y-OCC.

### 3.3 Video action recognition models

We study several existing video action recognition methods with a wide range of properties for this benchmark. We use CNN based R50[50], R2P1D[45], I3D[49] and X3D[12] and transformer based MViT[11], MViTv2[28] and VideoMAE[44] as our baseline models. This helps in making a comparison based on transformer and CNN based models. Further usage of lightweight models x3d allows us to compare the effect of model size to robustness to occlusion. Except for VideoMAE, all employed models underwent fully supervised training on the training split of the K-400 dataset. For the evaluation on UCF-101, all models were initialized with weights derived from the preliminary K-400 training phase. Sequentially, they were fine-tuned, on the training subset of the UCF-101 dataset. VideoMAE model underwent self-supervised training on K-400 and subsequently transitioning to fully supervised training on the K-400 training segment. On UCF-101, weights originating from fully supervised training on the K-400 training subset were employed. Augmented models were initialized with pre-trained K-400 weights. Following the introduction of synthetic occlusions to each video, each model was fine-tuned in a fully supervised manner using the UCF-101 training split.

### 3.4 Evaluation metric

We use accuracy and robustness scores as the metrics for evaluation. The robustness metric is defined in two ways, relative and absolute [38]. The absolute robustness score is given as $\gamma_p^a = 1 - \frac{A_c - A_p}{100}$ where $\gamma_p^a$ is the absolute robustness score for severity level p, $A_c$ is the accuracy on the clean dataset, and $A_p$ is the accuracy on the occluded dataset with severity level p. Relative robustness is defined as $\gamma_p^r = 1 - \frac{A_c - A_p}{A_c}$, where $\gamma_c^a$ is the relative robustness score.

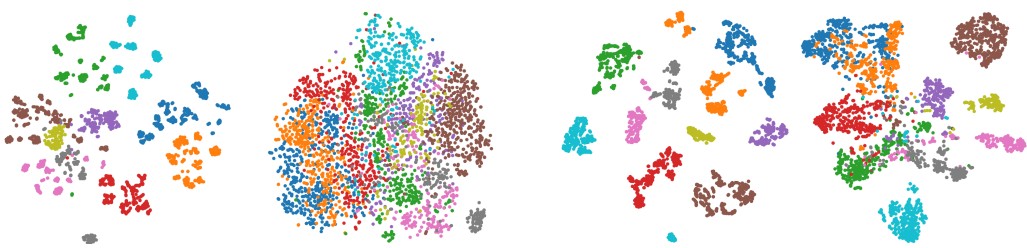

Figure 4: T-sne feature analysis under occlusion comparing CNN and transformer architecture. Left two plots: R2P1D features on UCF-101 and UCF-101-O respectively, and right two plots: MViT features on UCF-101 and UCF-101-O respectively.

### 3.5 Preliminary benchmark analysis

The evaluation of existing action recognition models on the proposed benchmarks is shown in Figure 3. We use clips of 224x224 for all datasets with 10 clips uniformly sampled across the temporal dimension, applying center crops to these clips. We observe that all the models show a drop in performance under occlusion, and the drop increases with the increase in severity. Our findings indicate a lack of generalization to occlusion in the existing baseline models. We also observe that transformer based models outperform CNN based models in both accuracy and robustness. This decline in accuracy with occlusion severity mirrors observations in the image domain [33] and video domain [38, 55], but the impact is more pronounced in videos.

We find Transformer-based models (MViT, VideoMAE, and MViTv2) exhibiting higher robustness to occlusion compared to R50, R2P1D, X3D, and I3D, which experience significant accuracy degradation with increasing occlusion severity. Additionally, in Table 1 we show accuracy results on real-world benchmark UCF-19-Y-OCC, revealing the lack of robustness in baseline models when exposed to natural occlusion. We analyze this further with the help of T-SNE plots (see Figure 4) and visualize the loss of discriminative power caused by occlusion in both CNN and transformers based models. These findings emphasize the need for future studies in action recognition to incorporate occlusion as a fundamental consideration.

## 4 Compositional model for robustness

We propose a simple transformer based compositional model which disentangles the occluder and activity features. Compositional models have been found effective in image domain [47] [22] and we extend this to videos. They act as efficient part detectors, where parts contribute to the overall class score of the region of interest. This enables the network to focus on visible parts and mitigate the impact of occlusion, resulting in robustness to occlusion in the image domain. The compositional net comprises three main components: *vmf kernels* $(\mu_1, \mu_2, \ldots, \mu_n)$, *occluder kernels* $(\beta_1, \beta_2, \ldots, \beta_n)$, and *class mixture models* $(A_1, A_2, \ldots, A_n)$. The primary objective is to obtain a generative model $p(F|y)$, where F represents the feature representation for a video and y represents the action label. The vmf kernels aid in identifying spatially relevant regions for the actions, while the class mixture model facilitates action classification by categorizing the features into different activities.

### 4.1 CTx-Net

The proposed CTx-Net takes a video input $X \in \mathbb{R}^{T \times H \times W}$, where $H$ is the height, $W$ is the width, and $T$ is the number of frames and outputs class probabilities. The model aims to disentangle occluded features from other features to mitigate their impact on the final class score. We build upon [22] by i) Employing a transformer backbone for modelling spatio temporal properties inherent to videos. ii) Usage of a collection of spatial and spatio temporal features for modelling occlusion.

**Video level representation** To effectively model temporal information, we utilize transformer architecture [46] as a backbone. The positional encoding preserves spatial information, while the transformer captures temporal dynamics. The output of the transformer network, denoted as $F_{tr} \in \mathbb{R}^{(H' \times W' \times T'+1) \times C'}$, represents the video's feature representation, where $H'$, $W'$, and $T'$ are the height, width, and temporal dimensions, and $C'$ is the number of feature dimensions. We combine

the feature embedding with the class embedding to integrate both information.We use MViT as the backbone in our model.

$$F_C = F_{tr} + cl^{encoding} \tag{1}$$

where $F_C \in R^{(H'W'T')\times C'}$ represents integrated features. Following this, we aggregate the temporal information by averaging across temporal dimension, thus obtaining a feature representation $F_l$ which contains both temporal and spatial information.

The feature map obtained from the transformer is then used to train the centers of the vmf cluster as described in [22]. The actions that are closest to the center of the cluster correspond to similar rudimentary actions, such as moving hands vertically.The mixture models are also learned in a manner similar to that presented in [22]. The mixture model learns actions movements which vote to compute the total class score for the entire action.

**Occluder kernel** The occluder kernel is obtained by clustering the vmf likelihoods. We explore occluder kernels for both temporal and spatio-temporal aspects. *Temporal kernel*: We obtain $F_c$ as described earlier. Subsequently, mean average pooling across the temporal dimension is applied to obtain $F_l$. The similarity between these features and the learned vmf kernels is computed to obtain $L_l$. We use features $l_p$ from $L_l$, where $F_l$ and $L_l \in \mathbb{R}^{(H'\times W')\times C'}$, for training the occluder kernels. *Spatial kernel*: We compute the similarity between features $F_c$ and the vmf kernels, resulting in $L_c$ where $F_c$ and $L_c \in \mathbb{R}^{(H'\times W'\times T')\times C'}$. From $L_c$, we sample the likelihood $l_c$, with dimensions $R^{C'}$. Spatial and temporal kernels are then combined to create spatio-temporal kernels. To train the occluder kernels, we utilize a set of random images that are unrelated to any specific action.

**End to end training** To incorporate the compositional model with a transformer, we use it as the classification head by replacing classical fully connected head. The differentiability of the model allows it to be fully trained by backpropagation. The trainable parameters are $T = \{\omega, \Lambda, A_y\}$. The loss function is similar to the one used in [22] and is defined as,

$$\mathcal{L}(y, y', F, T) = \mathcal{L}_{class}(y, y') + \gamma_1 \mathcal{L}_{weight}(\omega) + \gamma_2 \mathcal{L}_{vmf}(F, \Lambda) + \gamma_3 \mathcal{L}_{mixt}(F, A_y), \tag{2}$$

where $\mathcal{L}_{class}(y, y')$ is the cross entropy loss between network output and the target output $\mathcal{L}_{weight}(\omega)$ is the regularization term for the transformer network and $\mathcal{L}_{vmf}, \mathcal{L}_{vmf}$ regularize the parameters of the compositional head [48], where

$$\mathcal{L}_{vmf}(F, \Lambda) = -\sum_p \max_k log(p(f_p|\mu_k)), \tag{3}$$

$$\mathcal{L}_{mix}(F, A_y) = -\sum_p (1 - z_p)log[\sum_k \alpha_{p,k,y}^m p(f_p|\lambda_k)]. \tag{4}$$

Here $y$ is the predicted class score by the model $y'$ is the class label $\omega$ are the network weights.$\Lambda = \{\lambda_k = \{\sigma_k, \mu_k\}|k = 1\dots K\}$. $F$ is the feature map obtained from the transformer network, which is pooled across the temporal dimension. $\Lambda$ and $A_y$ are the parameters for the mixture model.

## 5 Experiments and analysis

**Training Setup** The parameters for compositional net have been trained in the same way as in [25]. We learn the parameters of n = 5 occluder models $\beta_1, ..., \beta_n$ in an unsupervised manner. We set the number of mixtures as 2. The mixing weights of the loss are determined empirically and are set to be $\gamma 1 = 0.1, \gamma 2 = 5, \gamma 3 = 1$. We train for 15 epochs using stochastic gradient descent [37] with momentum r = 0.9 and a learning rate of lr = 1e-4.

**Augmentation for robustness** Data augmentation provides a useful technique to solve distribution shift in the dataset and has been studied under various contexts [36]. We analyze its effectiveness for robustness against occlusion. We experimented with two different models, R2P1D and MViT, and use data augmentation by synthetically occluding video with random objects distinct from the objects used for benchmarking. We use linear motion with size varying between $20 - 40\%$.

### 5.1 Discussion and analysis

In Tables 1 and 2 we can observe that all the models suffer a performance drop under occlusion, with transformer based models performing better than CNN based models. Next, we analyze the effect of

Table 1: Comparison of performance on UCF101, UCF101-O and UCF-101Y-OCC datasets.

| Dataset | | UCF-101 | UCF-101-O | | | UCF-19-Y-OCC | | |
|---|---|---|---|---|---|---|---|---|
| Models | Aug | Top-1-Acc | Top-1-Acc | $\gamma^a$ | $\gamma^r$ | Top-1-Acc | $\gamma^a$ | $\gamma^r$ |
| R50 | × | 89.4 | 39.6 | 0.50 | 0.44 | 50.9 | 0.61 | 0.57 |
| R2P1D | × | 88.3 | 48.9 | 0.61 | 0.55 | 49.3 | 0.61 | 0.55 |
| X3D | × | 91.2 | 62.4 | 0.71 | 0.68 | 56.3 | 0.65 | 0.62 |
| I3D | × | 89.1 | 59.4 | 0.70 | 0.67 | 52.9 | 0.64 | 0.59 |
| MViT | × | 93.5 | 81.9 | 0.89 | 0.88 | 64.3 | 0.71 | 0.69 |
| VideoMAE | × | 96.0 | 79.2 | 0.83 | 0.82 | 65.4 | 0.69 | 0.68 |
| MViTv2 | × | 96.5 | 83.5 | 0.87 | 0.86 | 66.3 | 0.70 | 0.69 |
| R2P1D | ✓ | 83.0 | 71.4 | 0.89 | 0.86 | 37.5 | 0.55 | 0.45 |
| MViT | ✓ | 92.7 | 84.3 | 0.92 | 0.91 | 59.7 | 0.67 | 0.64 |
| MViTv2 | ✓ | 95.7 | 88.3 | 0.92 | 0.92 | 65.3 | 0.69 | 0.68 |
| VideoMAE | ✓ | 95.8 | 87.1 | 0.91 | 0.91 | 64.2 | 0.68 | 0.67 |
| CTx-Net | × | 93.4 | 86.4 | 0.93 | 0.92 | 67.4 | 0.74 | 0.72 |

Table 2: Comparison of accuracy and robustness score of studied models on the K-400-O dataset. CTx-Net2 uses MviTv2 backbone and CTx-Net-mae uses Video MAE as pretrained backbone.

| Dataset | K-400 | K-400-O | | |
|---|---|---|---|---|
| Model | Top-1-Acc | Top-1-Acc | $\gamma^a$ | $\gamma^r$ |
| R2P1D | 70.8 | 24.4 | 0.57 | 0.34 |
| R50 | 70.6 | 31.2 | 0.61 | 0.44 |
| I3D | 74.6 | 33.6 | 0.59 | 0.45 |
| X3D | 73.1 | 34.3 | 0.61 | 0.45 |
| MViT | 76.4 | 54.4 | 0.78 | 0.71 |
| MViTv2 | 77.6 | 56.7 | 0.79 | 0.73 |
| VideoMAE | 78.4 | 58.1 | 0.79 | 0.74 |
| CTx-Net | 75.9 | 58.4 | 0.79 | 0.77 |
| CTx-Net2 | 77.4 | 59.7 | 0.82 | 0.77 |
| CTx-Net-mae | 76.8 | 58.2 | 0.81 | 0.76 |

Table 3: Performance comparison showing accuracy with different severity levels and occluder motion on UCF-101-O dataset. Here L0 is 0%, L1 is (0-20)%, L2 is 20-40)%, and L3 is (40-60)% occlusion. S and D denote Static and Dynamic occluders respectively.

| Models | L0 | L1 -S | L1-D | L2-S | L2-D | L3-S | L3-D | Avg Acc |
|---|---|---|---|---|---|---|---|---|
| R50 | 89.4 | 84.7 | 52.9 | 71.3 | 34.3 | 46.7 | 16.5 | 56.5 |
| R2P1D | 88.3 | 79.3 | 45.3 | 55.3 | 27.2 | 33.3 | 19.3 | 49.7 |
| I3D | 89.1 | 85.2 | 56.7 | 71.7 | 40.2 | 46.8 | 23.7 | 59.1 |
| X3D | 90.3 | 85.3 | 46.6 | 73.5 | 34.4 | 47.0 | 21.8 | 56.9 |
| MViT | 93.5 | 91.8 | 86.6 | 87.2 | 80.3 | 80.2 | 70.1 | 84.2 |
| MViTv2 | 96.5 | 94.5 | 87.2 | 89.5 | 82.3 | 81.3 | 72.4 | 86.2 |
| VideoMAE | 96.0 | 94.7 | 87.8 | 91.2 | 80.1 | 81.5 | 66.7 | 85.4 |
| CTx-Net | 93.4 | 92.4 | 89.7 | 87.6 | 82.4 | 82.2 | 78.9 | 86.7 |

occlusion properties on the performance and also conduct controlled experiments in which we fix occlusion properties to study their effects.

**Effect of occluder dynamics** In Table 3 and 4 we show the impact of motion of occluders on models performance. We can observe most of the models excel with static occluders as compared with dynamic occluders. Table 4 demonstrates that both CNN-based and transformer-based models experience performance decline with increasing occluder motion complexity, particularly with random motion. This highlights the greater sensitivity of models to temporally incoherent motion compared to temporally coherent motion.

Table 4: Accuracy of models with occluders exhibiting different motions as well as occluders belonging to different classes; M1: straight line, M2: Random motion , M3: Circular Motion, M4: Static Motion, S1: Desktop, S2: Motorcycle, S3: Human, and S4: Cat. AA-S: average accuracy for shapes, and AA-M: average accuracy for motion variations.

| Models | S1 | S2 | S3 | S4 | AA-S | M1 | M2 | M3 | M4 | AA-M |
|---|---|---|---|---|---|---|---|---|---|---|
| R50 | 47.2 | 36.9 | 40.3 | 40.8 | 41.1 | 57.1 | 6.4 | 33.7 | 67.4 | 41.3 |
| R2P1D | 31.2 | 33.6 | 40.8 | 29.3 | 33.7 | 42.8 | 14.6 | 32.5 | 59.3 | 37.3 |
| I3D | 41.3 | 49.1 | 46.2 | 44.8 | 45.3 | 58.6 | 16.2 | 44.6 | 67.5 | 46.6 |
| X3D | 42.3 | 41.6 | 37.4 | 43.4 | 41.2 | 50.9 | 16.7 | 34.7 | 69.3 | 42.3 |
| MViT | 83.8 | 81.1 | 72.2 | 80.4 | 79.3 | 80.1 | 79.3 | 78.3 | 84.3 | 80.5 |
| MViTv2 | 85.8 | 83.7 | 75.2 | 82.1 | 82.6 | 83.9 | 77.3 | 81.3 | 88.3 | 82.7 |
| VideoMAE | 79.8 | 85.2 | 73.8 | 78.2 | 79.2 | 84.1 | 67.3 | 78.1 | 89.1 | 79.6 |
| CTx-Net | 87.7 | 85.2 | 74.3 | 83.2 | 82.6 | 85.2 | 81.2 | 84.3 | 86.3 | 84.2 |

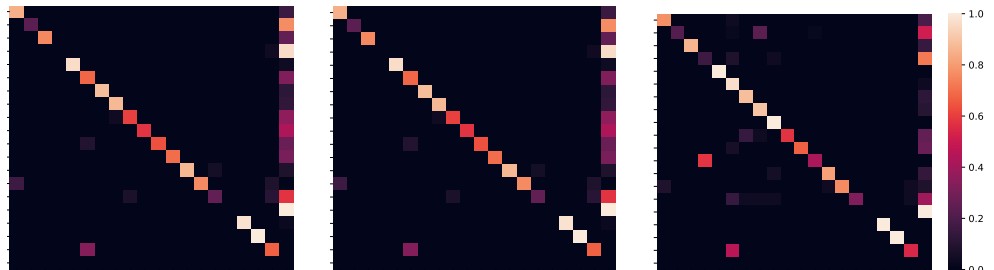

Figure 5: Class-wise performance analysis. Confusion matrix for (i) MViT, (ii) MViT+data augmentation, and (iii) CTx-Net on UCF-19-Y-OCC dataset.

**Effect of occlusion severity** To investigate the impact of occlusion severity, we vary the severity of the occluder from 0% to 60% (Figure 3 and Table 3). Notably, transformer-based models demonstrate significantly higher robustness to occlusion severity compared to CNN-based models.

**Effect of type of occluder** We vary the occluder's shape from spatially primitive (desktop) to spatially complex (human). We observe that baseline models perform poorly when the occluder is in the form of a human (Table 4). This pattern is also observed in CTx-Net, which consistently outperforms CNN-based and transformer based baselines across all occluders. Interestingly, X3D and R50 exhibit worse performance when the occluder is in the form of a motorcycle (less complex but more area).

**Effect of synthetic and natural occlusions** Comparing the occluded datasets UCF-101-O, K-400-O, and UCF-19-Y-OCC in Tables 1 2 and Figure 5, we observe that CNN-based models have low relative robustness scores for synthetic occlusions. However, for natural occlusions that appear intermittently in the video, CNN-based models exhibit higher relative robustness scores than in the case of synthetic occlusion. This suggests that CNN models are less resilient to spatially constant occluders than to occlusions that dynamically enter and exit the frames. On the other hand, transformer based models demonstrate a larger decrease in relative robustness scores for natural occlusions that sporadically appear throughout the video, as opposed to synthetic occlusions that persist throughout. This indicates that transformer-based models are more susceptible to dynamic natural occlusions than to spatially constant occlusions. More specifically in UCF-101-O, a dataset with synthetic augmentation, synthetic occluders cause a distribution shift that is constant across training and test sets which results in increased robustness for data augmented methods. However, this is not the case for UCF-19-Y-OCC where augmented models are less robust, indicating a difference in distribution shift caused by natural and synthetic occlusions.

**Effect of pretraining** Next, we study the effect of pretraining on robustness. We experiment with both CNN and transformer based models.To test this we train models from scratch, on UCF-101 dataset and compare their performance with models finetuned on UCF-101 dataset, these models weights were initialized with pretrained weights on K-400 dataset. Figure 6 shows that pretraining clearly helps in increasing the performance of models and making it more robust to occlusion. Furthermore, pretraining increases relative robustness of transformer based models more than CNN based models.

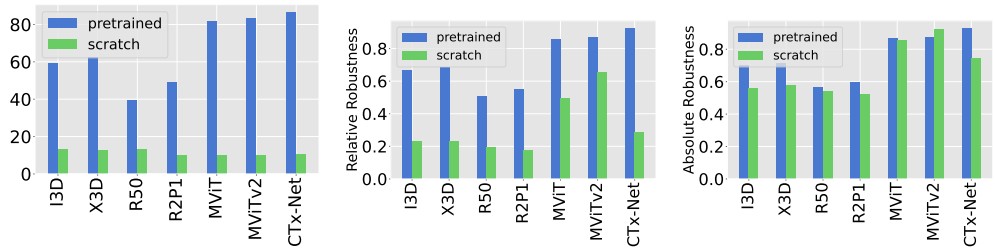

Figure 6: Effect of pretrained weights on models performance for UCF-101-O dataset. Left: accuracy, middle: relative robustness $\gamma^r$, and right: absolute robustness $\gamma^a$.

Table 5: Ablations to study the effect of different components of CTx-Net on UCF101-O dataset.

| Dataset | UCF-101 | UCF-101-O | | |
|---|---|---|---|---|
| Models | Top-1-Acc | Top-1-Acc | $\gamma^a$ | $\gamma^r$ |
| CTx-Net (CNN) | 24.2 | 10.3 | 0.86 | 0.43 |
| CTx-Net (spatial occluder kernels) | 90.3 | 84.2 | 0.94 | 0.93 |
| CTx-Net (w/o class encoding) | 91.2 | 83.3 | 0.92 | 0.91 |
| CTx-Net (pretrained with occluded data) | 85.2 | 75.4 | 0.89 | 0.88 |
| CTx-Net | 93.4 | 86.4 | 0.93 | 0.92 |

## 5.2 Improving robustness against occlusion

We observe that augmentation does help with robustness against occlusion in synthetic benchmarks for both transformer and CNN based models but fails to generalize to natural occlusion (Table 1). The proposed CTx-Net, a compositional model, improves upon the performance on synthetic occlusion as well outperforms all the models on natural occlusion.

**Ablations on CTx-Net** We perform ablations of the proposed CTx-Net on UCF-101-O dataset to demonstrate the effects of the chosen components on its robustness to occlusion and report its results in Table 5. *CNN vs transformer backbone:* First, we compare the effect of CNN vs transformer backbones on the network. We observe that CNN based network is not able to model long dependencies, which are essential. *Spatial vs spatio-temporal kernels:* We perform experiments with spatial occluder kernel as opposed to the spatio-temporal occluder kernel. We observe a drop in performance, demonstrating the importance of spatio-temporal nature of occlusion. *Impact of class token:* We also demonstrate the importance of using class encoding during classification using CTx-Net. As we can see from Table 5 the class encoding captures information from all patches to determine the class as in [51], [52]. *Training on occluded data:* We also analyze the performance of CTx-Net when trained using occluded data. We can clearly see that there is a significant drop in performance as compared to vanilla CTx-Net. This can be attributed to the presence of occluders causing the class mixture model to treat occluder as an essential property for action detected.

## 6 Conclusion and findings

In this work, we focus on video action recognition under occlusion. To the best of our knowledge, this is the first study focusing on this problem. To study this problem, we propose two synthetic benchmarks datasets based on UCF-101 and Kinetics-400 datasets and also a dataset containing naturally occluded videos. We observe several interesting findings such as 1) *transformers more robust than CNNs*, 2) *pretraining helps all models*, and 3) *augmentation helps, but does not generalize to real-world occlusions*. We also propose a simple transformers based compositional model which outperforms existing methods and even augmentation based models on synthetic benchmarks as well as on the proposed naturally occluded benchmark. All three benchmark datasets and code is publicly available at this [link].

# 7 Acknowledgements

This research is based upon work supported in part by the Office of the Director of National Intelligence (Intelligence Advanced Research Projects Activity) via 2022-21102100001 and in part by University of Central Florida seed funding. The views and conclusions contained herein are those of the authors and should not be interpreted as necessarily representing the official policies, either expressed or implied, of ODNI, IARPA, or the US Government. The US Government is authorized to reproduce and distribute reprints for governmental purposes notwithstanding any copyright annotation therein.

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

# A   Appendix

## A.1   Overview

All three benchmark datasets and code will be made publicly available at this [link]. We include the following results and details in this supplementary,

1. We provide classwise performance of CTx-Net on UCF-101 and UCF-101-O dataset.
2. We provide additional results for data augmented transformer.
3. We provide a detailed network architecture of the proposed CTx-Net.
4. Qualitative Results for CTx-Net.
5. We provide tsne visualizations for effect of occlusion.
6. We further provide the confusion matrix on UCF-101-Y-OCC dataset for other models as well.
7. Limitation and Social Impact.
8. Datasheet for the proposed datasets.

## A.2   Class wise Performance

In figure 7 we provide the classwise performance for the proposed CTx-Net on UCF-101 and UCF-101-O dataset. The result shows the relative robustness score for each class. As expected for most of the classes, the relative robustness score is less than 1 which shows the negative effect of occlusion on the task of action recognition. Further, a few outlying classes have robustness score more than 1 which shows that the model performs better on occluded frame as compared to clean frames. We can also notice from the plots that for most of the classes the score is close to 1 which shows the robustness of the proposed CTx-Net irrespective of classes.

## A.3   Additional Results

In Figure 13 performance of MVit+data augmentation on out of distribution occluders is shown. It is performed by including a specific number of occluders during test time which were present during training as well. From the Figure we can clearly see that the proposed CTx-Net has minimal effects on performance while varying the distribution of occluders whereas the performance of data augmented model falls as the number of out of distribution occluders increase. This shows data augmentations inability to generalize well.

## A.4   UCF-19-Y-OCC dataset

The UCF-19-Y-OCC is composed of 19 classes. The classes included are - Band Marching, Bench-Press, Biking, Playing Cello, Baby Crawling, Walking with dog, Drumming, Playing Flute, Hand stand Pushups, Kayaking, Mopping Floor, Nunchucks, Pizza Tossing, Pushups, Skateboarding, Skiing, Soccer Juggling, Soccer Penalty, Surfing. All of these actions are also present in UCF-101 dataset. Figure 14 shows the distribution of number of clips per class in the dataset. Figure 13 shows the confidence matrix plot for I3D and X3D on UCF-19-Y-OCC dataset. It can be clearly seen that these methods do not have enough discriminative properties in case of natural occlusion.

## A.5   Effects of Occlusion on feature representation

From Figure 12 we can see that the feature representations learned by X3D and I3D are quite discriminative in case of clean frames, as they are able to provide distinct cluster in the tsne plot, whereas for UCF-101-O we can see that the feature representation does not have enough discriminative power given the lack of distinct clusters.

## A.6   Qualitative Results

Figure 15 shows qualitatively the localization of occluders in a video. Each row represents frames in a video, followed by localization of occluders performed by CTx-Net. We observe that CTx-Net

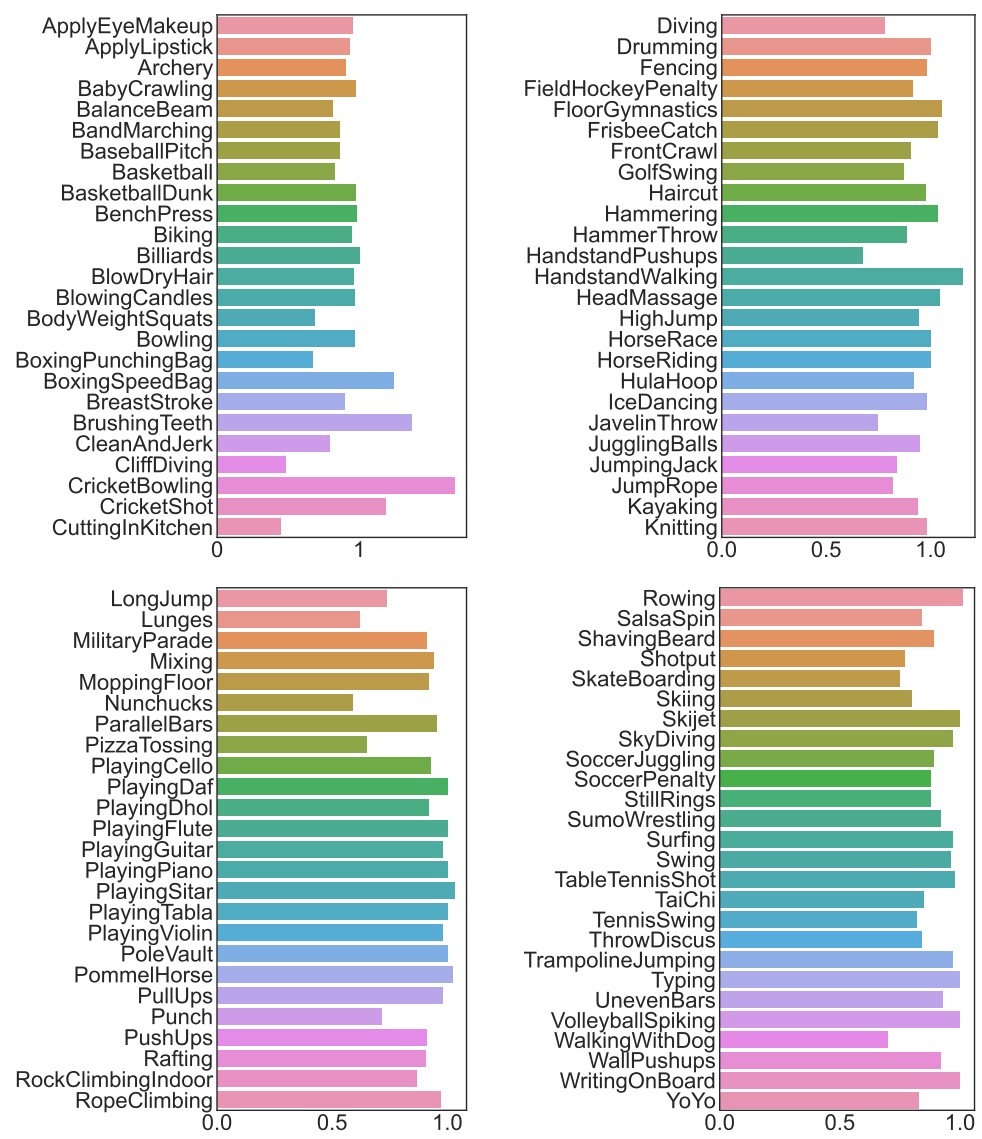

Figure 7: Class wise performance of CTx-Net on UCF-101 dataset using relative robustness as the metric.

is able to localize occluders exhibiting a variety of shapes and motions effectively. Additionally, we present qualitative findings pertaining to the CTx-Net model trained on data augmented with synthetic occlusions, as shown in Figure 16. Notably, we observe an interesting phenomenon: the model tends to identify certain attributes as occlusions even within video segments that remain unoccluded throughout. This behavior is rooted in the utilization of augmented data to train a class mixture model within the CTx-Net architecture. Consequently, this affects the model's ability to generalize effectively to non-occluded scenarios. Our study also extends to the comparative analysis detailed in Table 3 and Table 4. Particularly, we analyze instances where VideoMAE's performance is lower in contrast to MViTv2 and MViT. Visualizations of videos that exhibit correct classification by MViT and MViTv2, yet are misclassified by VideoMAE, are shown in Figure 17. These observations contribute to a comprehensive understanding of the strengths and limitations exhibited by the models under study.

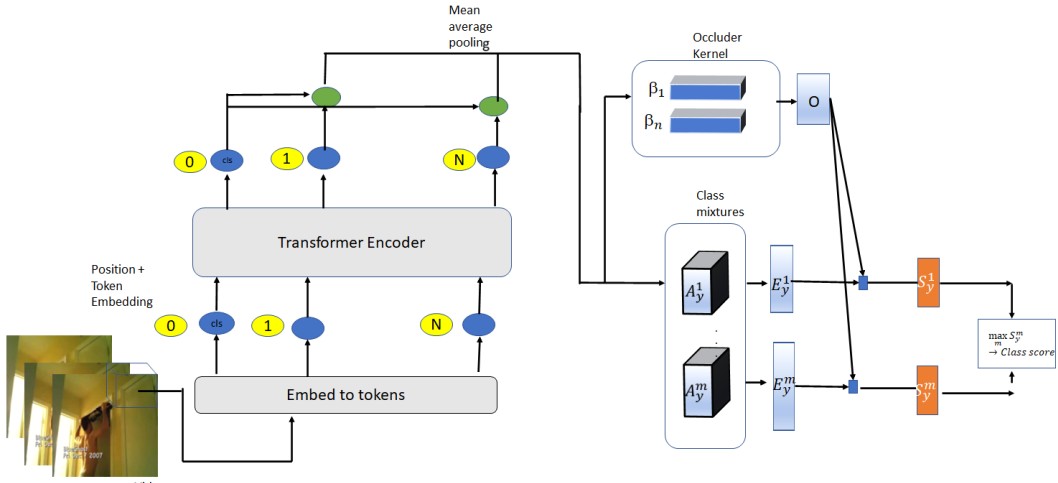

Figure 8: Overview of the CTx-Net architecture. A transformer network used to extract features .Class token is then aggregated with feature tokens, followed by feature pooling using average pooling, followed by this a convolution with vMF kernels $\mu_n$ followed by non-linear vMF activation $\mathcal{N}(.)$.The resulting vMF likelihood L is used to compute the occlusion likelihood O using the occluder kernels $\beta$.Furthermore, L is used to compute the mixture likelihoods $E_y^m$ using mixture models $A_y^m$. O and $E_y^m$ compete in explaining L the orange box and are combined to obtain the final class score.

c

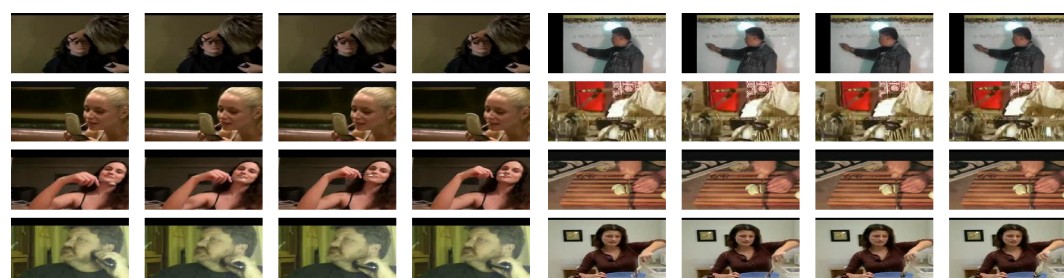

Figure 9: Visualization for the patches in a video activated by three different vmf kernel. Left column: represents actions which comprises hand movements closer to upper body, right column: represents action which comprises hand movements closer to lower body

## A.7 Model architecture

Figure 8 shows the model architecture of the proposed CTx-Net which uses a transformer backbone. To calculate the class score, first the feature and class tokens are obtained for the given video. Following this, the feature token and class tokens are aggregated. Then vmf likelihood of the obtained features is then calculated. Class models are used to obtain class likelihood for each part detected, Similarly the occluder model is then used to obtain likelihood of which fea- tures are occluded.Both

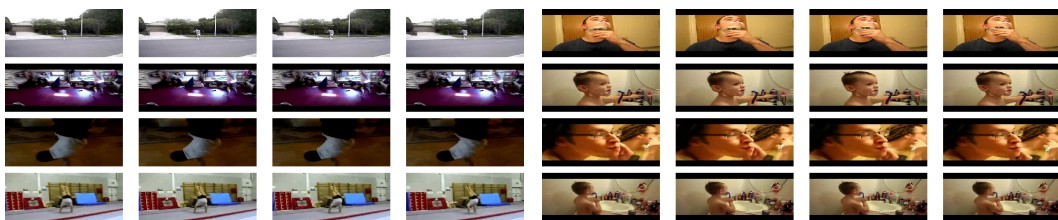

Figure 10: Visualization for the patches in a video activated by different components of mixture model for two different classes.

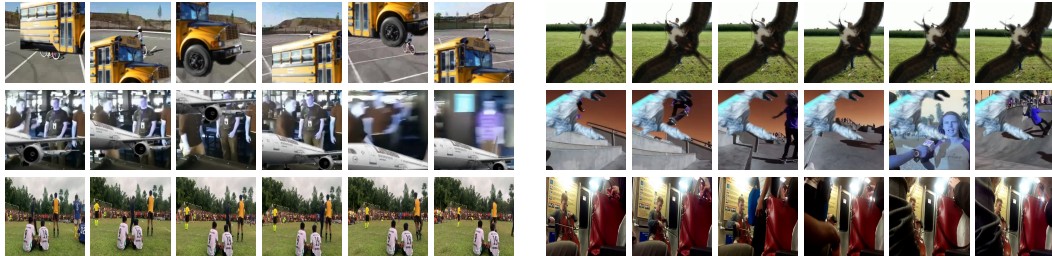

Figure 11: Samples from occluded benchmark datasets. First row: UCF101-O, second row: K-400-O, and third row UCF-19-Y-OCC.

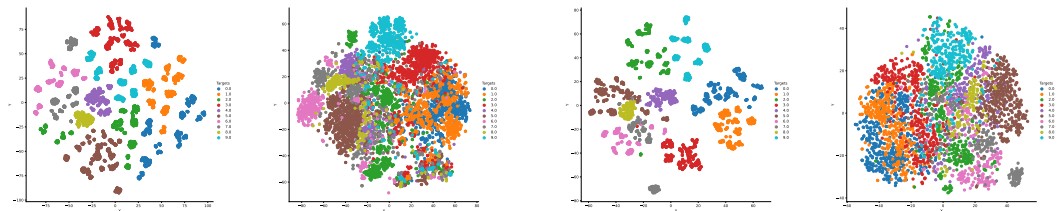

Figure 12: T-sne analysis of features under occlusion comparing I3D and X3D. Starting from left, (i) I3D features on UCF-101, (ii) I3D features on UCF-101-O, (iii), X3D features on UCF-101 and (iv) X3D features on UCF-101-O.

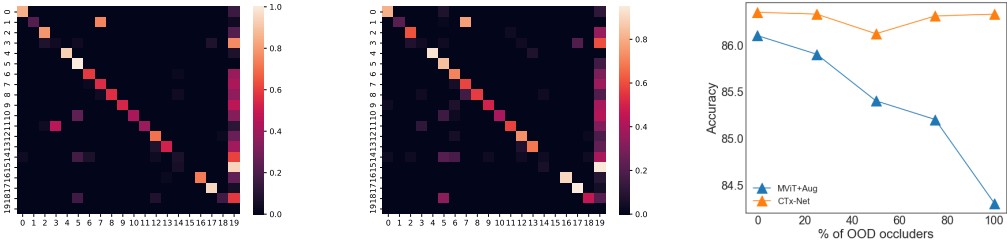

Figure 13: Class-wise performance analysis. Confusion matrix for (a) I3d, and (b) X3d. (c)Performance of the proposed CTx-Net and MViT+data aug on out of distribution occluders.

these scores are then combined to obtain the final class score. We also visualize the patches which activate the vmf and class mixture models the most. In Figures 9 and 10 we can see that the vmf kernels learned corresponds to more fundamental movements like moving the hand up, whereas class mixture model capture same actions being performed from differing points of views.

## A.8 Occluders

From Figure 18 we can see some of the images that were used for training the occluder kernel. These are a randomly selected, out of distribution images in which no action seems to take place. Hence, this helps in separating out the random occlusion that occur in the video. Figure 11 also shows the occluded samples from the proposed datasets.UCF-101-O and K-400-O showing different severity of occlusions used

## A.9 Limitations and Societal Impact

Benchmarking computer vision models for occlusion-aware video action recognition can lead to significant advancements in various application domains like development of enhanced surveillance systems which take occlusion into account, autonomous driving among others. These enhanced surveillance systems might cause some privacy concerns. For synthetically occluded datasets since, we sample objects randomly from the PASCAL VOC dataset the occluder can often present an unrealistic appearance both since the texture of occluder is significantly different from that of rest

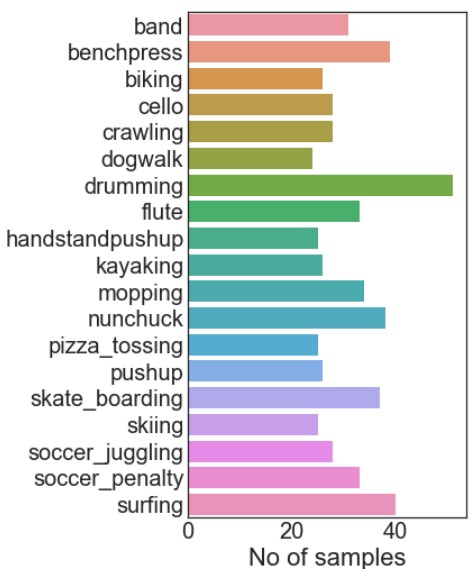

Figure 14: Class wise distribution of samples in UCF-19-Y-OCC dataset.

of the scene but also, the motion of exhibited by the objects is relatively simple as opposed to the complicated motions followed real life occluders.

### A.10 Datasheet for Dataset

**Motivation** The core motivation behind this dataset was to study the effect of natural occlusion on video action recognition models, which could further help in moving the field forward by studying the performance of models systematically and in a real world setting.

**Composition** **Content and Composition** The instances in the dataset consists of videos in which actions are completely or partially occluded. The UCF-101-Y-OCC dataset consists of 19 classes, each class consists of 30 clips each lasting 5 second. The UCF-101-O and K-400-O consists of all the videos in test split of UCF and Kinetics dataset occluded synthetically.

**Does the dataset contain all possible instances or is it a sample (not necessarily random) of instances from a larger set?** Yes for K-400-O and UCF-101-O, No for UCF-101-Y-OCC.

**Are relationships between individual instances made explicit (e.g., users' movie ratings, social network links)?** No.

**Are there recommended data splits (e.g., training, development/validation, testing)?** No, all the proposed datasets are for evaluation only.

**Are there any errors, sources of noise, or redundancies in the dataset?** No, The entire UCF-101-Y-OCC dataset was annotated by the authors.

**Is the dataset self-contained, or does it link to or otherwise rely on external resources (e.g., websites, tweets, other datasets)?** UCF-101-Y-OCC is composed of YouTube videos. UCF-101-O and K-400-O are self-contained.

**Is there a label or target associated with each instance?** Yes, a class label representing the action is associated with each dataset.

**Does the dataset contain data that might be considered sensitive in any way (e.g., data that reveals race or ethnic origins, sexual orientations, religious beliefs, political opinions or union memberships, or locations; financial or health data; biometric or genetic data; forms of**

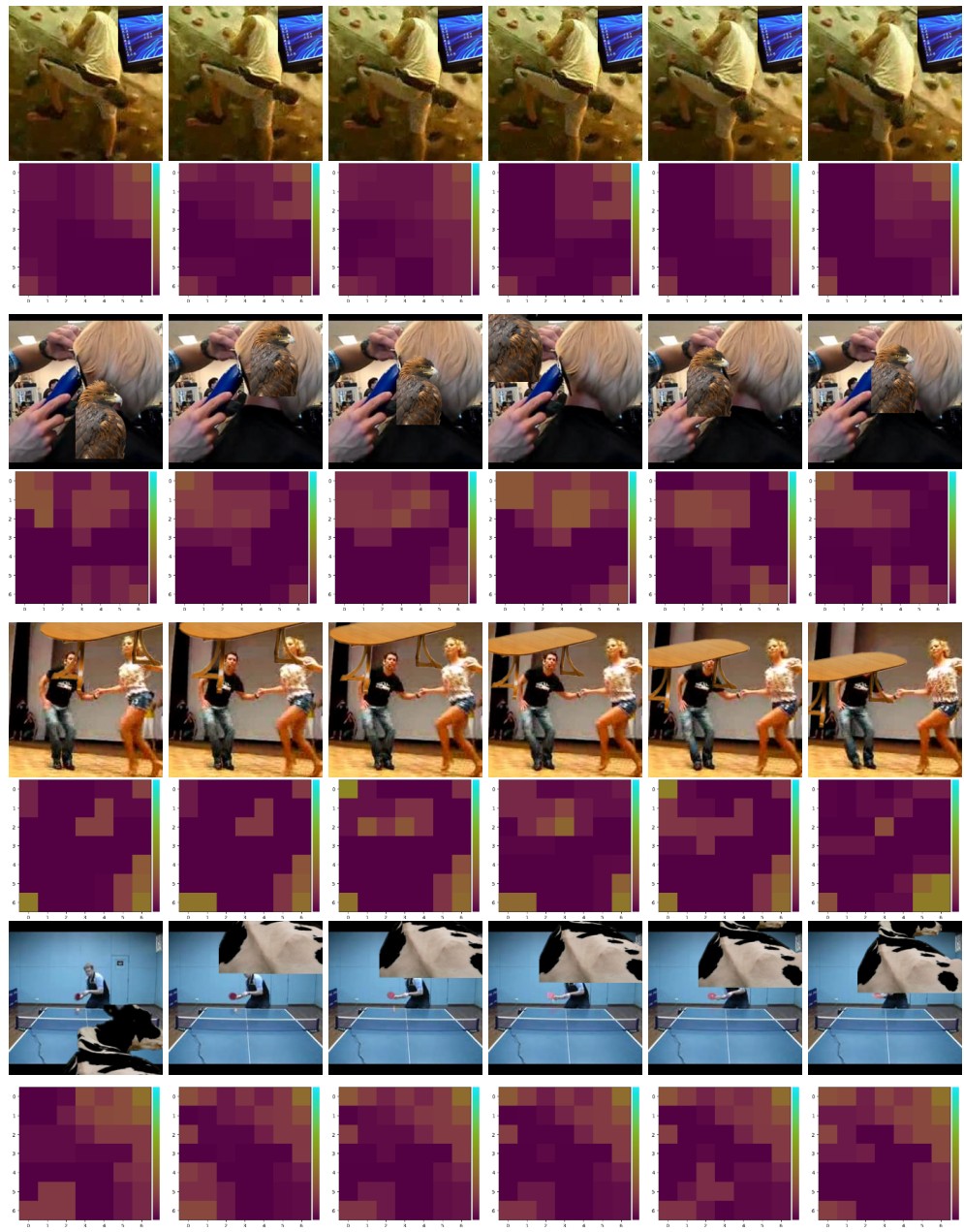

Figure 15: Qualitative results for occlusion localization. Each row represents a video followed by localization of occlusion in each of the above frames.

government identification, such as social security numbers; criminal history)?  No.

**Does the dataset contain data that might be considered confidential (e.g., data that is protected by legal privilege or by doctor–patient confidentiality, data that includes the content of individuals' non-public communications)?** No.

**Does the dataset contain data that, if viewed directly, might be offensive, insulting, threatening, or might otherwise cause anxiety?** No.

**Does the dataset identify any subpopulations (e.g., by age, gender)** No.

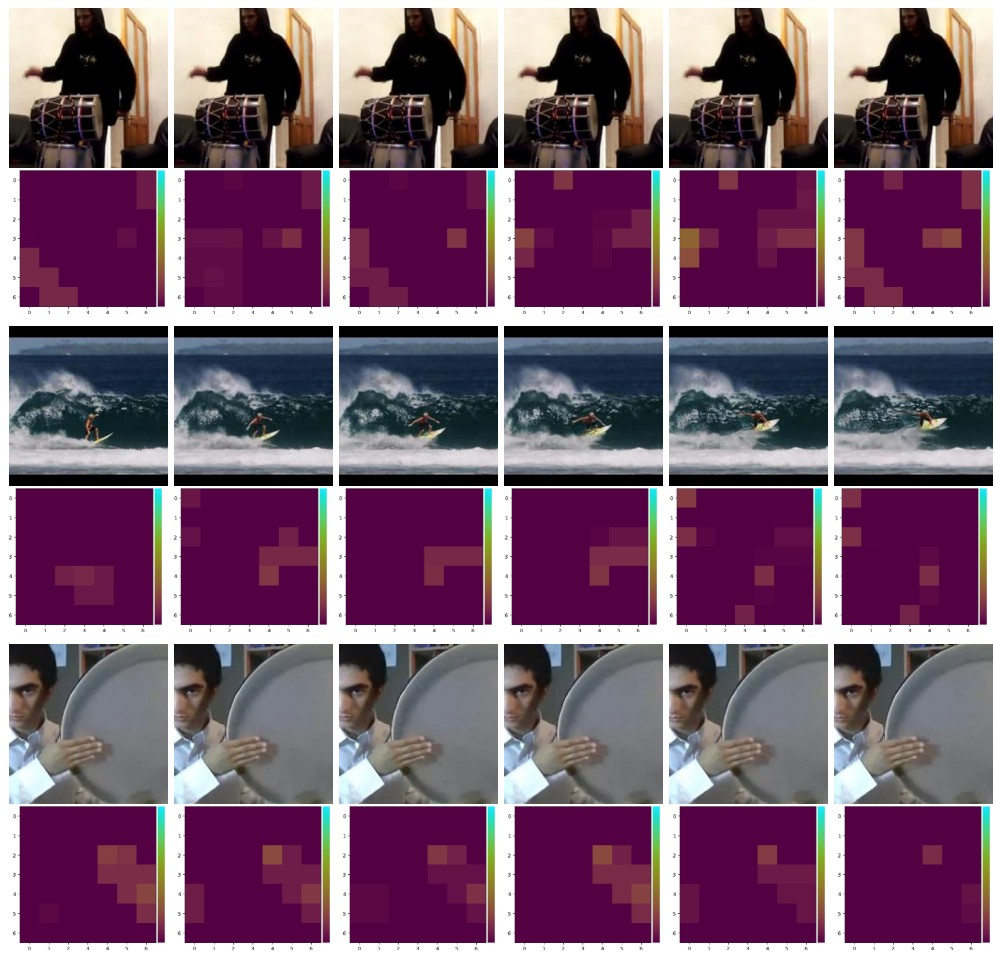

Figure 16: Qualitative results for occlusion localization of CTx-Net (augmented). Each row represents a video followed by localization of occlusion in each of the above frames.

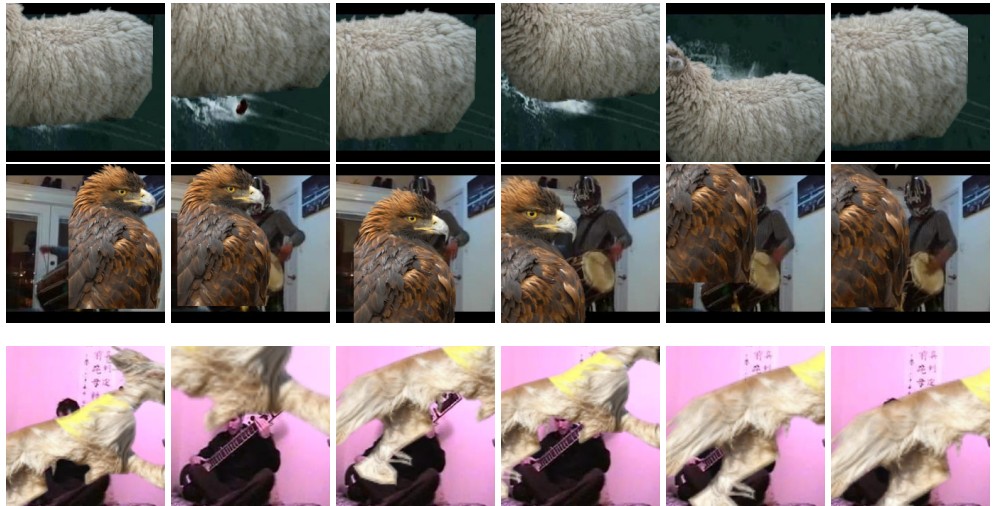

Figure 17: Examples of videos in which Video MAE is unable to classify correctly whereas MViT and MViTv2 are able to correctly classify.

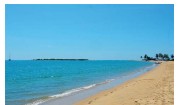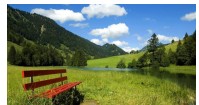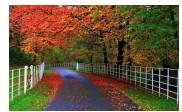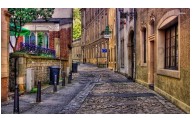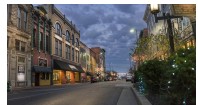

Figure 18: Visualization of images used for training occluder model

**Is it possible to identify individuals (i.e., one or more natural persons), either directly or indirectly (i.e., in combination with other data) from the dataset?** Yes, the individual might be identifiable from YouTube.

**Collection process** Each instance of UCF-101-Y-OCC was extracted from publically available videos on YouTube. The entire video was then viewed to ensure the action occurring is relevant as well as a part of it is occluded. UCF-101-O and K-400-O are composed of test split of UCF-101 and K-400 datasets.

**What mechanisms or procedures were used to collect the data (e.g., hardware apparatuses or sensors, manual human curation, software programs, software APIs)?** YouTube API was the only external source of data used for UCF-101-Y-OCC. Annotations for UCF-101-Y-OCC were performed manually by the authors.

**If the dataset is a sample from a larger set, what was the sampling strategy (e.g., deterministic, probabilistic with specific sampling probabilities)?** Not applicable since UCF-101-O and K-400-O contain entire test split of UCF-101-O and K-400-O.

**Over what timeframe was the data collected?** 2 Months for UCF-101-Y-OCC.

**Did you collect the data from the individuals in question directly, or obtain it via third parties or other sources (e.g., websites)?** YouTube was used for UCF-101-Y-OCC. UCF-101 and K-400 were synthetically occluded to obtain UCF-101-O and K-400-O.

**Were the individuals in question notified about the data collection?** Data was publically available on YouTube.

**Labelling/Preprocessing/cleaning** The video collected from YouTube was the segmented temporally into 5 seconds segments which contained occlusion. UCF-101-O and K-400-O have annotations same as UCF-101 and K-400 whereas UCF-101-Y-OCC was annotated manually.

**Uses** **Has the dataset been used for any tasks already?** For testing the performance of current models when exposed to occluded actions.

**Is there a repository that links to any or all papers or systems that use the dataset?** NA

**What (other) tasks could the dataset be used for?** Action Recognition under occlusion is the only intended task for the proposed dataset.

**Is there anything about the composition of the dataset or the way it was collected and preprocessed/cleaned/labeled that might impact future uses** No.

**Are there tasks for which the dataset should not be used?** No.

**Distribution** **Will the dataset be distributed to third parties outside the entity (e.g., company, institution, organization) on behalf of which the dataset was created?** No

**How will the dataset be distributed (e.g., tarball on website, API, GitHub)?** Will be available along with the code through download link.

**When will the dataset be distributed?** During the review process.

**Will the dataset be distributed under a copyright or other intellectual property (IP) license, and/or under applicable terms of use (ToU)?** Dataset will be distributed under Creative Commons License.

**Have any third parties imposed IP-based or other restrictions on the data associated with the instances?** No

**Do any export controls or other regulatory restrictions apply to the dataset or to individual instances?** No

