# OpenReview forum: "Revealing the unseen: Benchmarking video action recognition under occlusion"
_NeurIPS.cc/2023/Track/Datasets_and_Benchmarks — NeurIPS 2023 Datasets and Benchmarks Poster_

### Official Review · Reviewer_AuqS · 2023-07-18
**An Action Recognition Benchmark using Videos with Occlusions**

**Rating:** 6
**Confidence:** 3
**Correctness:** Most claims made in the submission ar…
**Clarity:** Yes, the paper is well written.

**Strengths:**

1. This paper collects a real-world occlusion dataset, which is beneficial to the study of video analysis towards occlusion.
2. This paper provides detailed experiments to evaluate the performance of different models regarding different severities of occlusion.

**Additional Feedback:**

No additional feedback.

**Documentation:**

Most of the materials are about the models and their performance. It seems that the documents to describe the dataset and benchmark are not enough.

**Ethics:**

No ethics issues.

**Limitations:**

1. The description of data collection/generation is not detailed enough.
2. In Line 26, “UFC-101-O” should be “UCF-101-O”.
3. Generally, this paper looks like proposing a new model/algorithm (i.e., CTx-Net), rather than providing a detailed dataset or benchmark, making the paper not match this track well.

**Opportunities For Improvement:**

A proportion of the paper describes the proposed CTx-Net. However, given that this is a dataset and benchmarking track, the authors can provide more details about data collection, rather than the proposed model.

**Relation To Prior Work:**

It seems that there are also other papers about occlusions in video classification. The authors have not clearly discussed them.

**Summary And Contributions:**

This paper discusses the occlusion issues of video action recognition. The authors proposed three datasets, two of which are synthesized with existing datasets, one of which is collected in the real world. The authors further proposed a model, namely CTx-Net, which outperformed existing models on action recognition under occlusion. Generally, this work explores different video action recognition models towards occlusions.

---

> ### Author Response · Authors · 2023-08-22
>
> We thank the reviewer for their time, insightful comments, and questions. We have provided our responses below.
>
> 1.1 Dataset collection and visualization for dataset \
> We have added more dataset collection details, detailing the criteria for selection of videos and the processing of the videos used for evaluation for UCF-19-Y-OCC dataset in section . We have further added visualizations for UCF-101-O and K-400-O dataset.
>
> 1.2  Change Line 126 from UFC-101-O to UCF-101-O \
> Thanks for pointing this out. We have made these changes in the updated version
>
> 1.3  Problem of paper as model proposal rather than a benchmarking one. \
> Thank you for your observation regarding the perceived emphasis on the proposed model, CTx-Net. In response to your feedback, we've undertaken several measures to ensure a more balanced presentation:
> * We've provided comprehensive insights into the dataset collection process, including video selection criteria and video processing for UCF-19-Y-OCC dataset. This information is available in Section 3.2 .
> * To enhance the dataset context, we've included visualizations for UCF-101-O and K-400-O datasets, offering a clearer understanding of the underlying data.
> * We've introduced augmented results for MViTv2 and VideoMAE in Table 1, along with outcomes for MViTv2 and Video in Tables 3 and 4, enhancing the benchmarking analysis.
> * To provide a more comprehensive view, we've incorporated CTx-Net results for situations where the backbone is trained from scratch on UCF-101 in Figure 6.
> These changes aim to strike a better balance between the proposed model and the benchmarking aspect, ensuring that the paper aligns more effectively with its intended track. Your insights have been invaluable in guiding these enhancements.
>
> 2.1 Relevant literature \
> We have included relevant works and updated Section 2 to reflect these changes.
>
> 3.1 Lack of documentation benchmark and dataset.\
>  Hopefully the aforementioned added details along with the updated submission would be able  to strike a better balance between the proposed model and the benchmarking aspect.

---

> > ### Author Response · Authors · 2023-08-29
> >
> > We hope we have addressed all the concerns raised by the reviewer. If there is still any open issue, we will do our best to resolve it.
> >
> > Once again, we would like to thank the reviewer for the time and effort in reviewing our work.

---

> > > ### Comment · Reviewer_AuqS · 2023-08-30
> > > **Raise my ratings from 5 to 6**
> > >
> > > Dear authors,
> > >
> > > I greatly appreciate your responses and hard work on paper revision. Most of my concerns have been resolved. I raise my ratings from 5 to 6. Thank you so much.
> > >
> > > Best wishes,
> > >
> > > Reviewer AuqS

---

### Official Review · Reviewer_EmRM · 2023-07-21
**Interesting benchmarking study on occluded video classification but lacks some key points**

**Rating:** 6
**Confidence:** 4
**Correctness:** As mentioned, the claims about pre-tr…

**Strengths:**

1. It constructs the benchmark datasets with systematically designed occlusions. Besides the synthetic occlusion, a natural occlusion dataset is also collected from Youtube. It makes the benchmarking results more convincing with real occlusion video.

2. It proposes a novel CTxNet based on a compositional network, to address the occlusion issue. From the experimental result, there is improvements in robustness, compared to the baseline MViT.

3. The benchmarking study provides several interesting findings. The transformers degrade more significantly on natural occlusion while the CNNs are more resilient; the shape of occlusion has large impact on the performance of the models.

Overall, it is a complete benchmarking work in video classification, from dataset construction, then benchmarking study, next ablation study, finally propose novel technique to address the issue.

**Additional Feedback:**

N.A

**Clarity:**

The paper is ok to follow. However, it mixes the benchmarking study with the proposed technique, the pages for experimental analysis are quite limited.

**Documentation:**

The documentations look fine.

**Ethics:**

N.A

**Limitations:**

Same as above.

**Opportunities For Improvement:**

1. Though this work emphasizes the finding that transformers are more robust than CNNs, previous work[1] has claimed these findings in video classification robustness and it is not cited there.

2. CTxNet has marginal improvement on MViT/MViTv2 from Table 1. It has at least 3 weights to balance different optimization objects, making it hard to train the model and find proper weights.

3. Though it claims that pre-trained model can improve robustness in the abstract, there is only a small paragraph mentioning it in section 5.1. Moreover, the setting of pretraining is not available in the whole paper. I would like to see more details for the pretraining part if it is so effective.

[1]Yi, Chenyu, et al. "Benchmarking the robustness of spatial-temporal models against corruptions." arXiv preprint arXiv:2110.06513 (2021).

**Relation To Prior Work:**

As mentioned in the limitations.

**Summary And Contributions:**

This work proposes 3 occluded video classification datasets, UCF101-O, Kinetic-400-O and UCF101-Y-OCC. It introduces different occlusions with various occlusion types, severity and motion modes. Based on the datasets, it conducts a benchmarking study to evaluate the robustness of existing CNN and transformers.  Besides, it proposes a novel method named CTx-Net to alleviate the degradation of performance when the model encounters occlusion in the video. Finally, it founds some insights in terms of model, training and data augmentation.

---

> ### Author Response · Authors · 2023-08-22
>
> We thank the reviewer for their time, insightful comments, and questions. We have provided our responses below.
> 1.1 Missing Literature \
> Thanks for mentioning this relevant paper. We have included this work in the recent works as well as the discussion sections.
>
> 1.2 CTx-Net has marginal improvement on MViT/MViTv2 from Table 1. \
> This observation regarding CTx-Net's performance relative to MViT/MViTv2 in Table 1 is indeed perceptive. While the enhancement in performance is not huge for UCF-101-O, it becomes particularly significant in the context of UCF-19-Y-OCC, which mimics real-world conditions.CTx-Net employs an MViT backbone, a component that remains fixed during training. It's important to highlight that this backbone is adaptable and can be substituted with alternative backbones as needed. In Table 2, we observe that both MViTv2 and Video MAE backbones exhibit an upward trend in performance compared to the base CTx-Net configuration, further affirming the adaptability and potential for improvement within the CTx-Net framework.
>
> 1.3  Details regarding pretraining \
> We sincerely appreciate the reviewer's concern regarding the training procedures of the studied models. Except for VideoMAE, all the studied models use fully  supervised training on the training split of K-400 dataset. To obtain results on UCF-101, we initialized all models with weights acquired from the K-400 training phase. Subsequently, fine-tuning was carried out in a fully supervised manner on the training set of the UCF-101 dataset. VideoM.AE was subject to additional self-supervised training on K-400 followed by supervised training on the training split of K-400. For UCF-101, weights garnered from the fully supervised training on the K-400 training set are used for the fine-tuning process. In the case of augmented models, initialization occurred with pre-trained K-400 weights. After this, synthetic occlusion is used on video samples as augmentation for fine tuning the model on  UCF-101 training split. More details about augmentations are provided in section 5 of the submission.  We acknowledge that the clarity of this information could benefit from improvement and have addressed it accordingly in the updated submission.
>
> 2.1 Added Experiments to address the concern for limited analysis and experiments\
> Thank you for your feedback on the structure of the paper. We've taken your input into consideration and made the following modifications:
>
> * We have introduced augmented results for MViTv2 and VideoMAE in Table 1, and incorporated outcomes for MViTv2 and Video in Tables 3 and 4 to provide a more comprehensive benchmarking analysis.
>
> *  To offer a more well-rounded perspective, we've included results for CTx-Net in scenarios where the backbone is trained from scratch on UCF-101.
>
> *  Recognizing the need for clearer illustration, we've augmented the paper with additional dataset visuals to provide a more comprehensive understanding.
>
> These modifications aim to address the balance between benchmarking and the proposed technique, as well as enhance the overall clarity and depth of the experimental analysis. Your insights have been invaluable in shaping these improvements.

---

> > ### Comment · Reviewer_EmRM · 2023-08-27
> > **Questions addressed**
> >
> > Thank you for the modifications and improvements. The added experimental results and analysis addressed my concerns, especially on pre-training. It is a good paper that contributes a comprehensive overview of the impact of occlusion in video classification. Considering the contribution is incremental, I will raise the score to 6, as a borderline paper.

---

> > > ### Author Response · Authors · 2023-08-29
> > >
> > > We thank the reviewer for their valuable time and effort in reading our response. We are glad that we were able to address all the queries.
> > >
> > > Regarding the comment 'contribution is incremental', we hope it is for the new baseline we have proposed to study this novel problem. To best to four knowledge, there is no existing work focusing on occlusion for video action recognition and we propose three novel datasets in this benchmark to study this problem.
> > >
> > > Once again, we truly appreciate the reviewers' effort in this reviewing process.

---

### Official Review · Reviewer_pkpL · 2023-07-22
**Review of Revealing the unseen: Benchmarking video action recognition under occlusion**

**Rating:** 6
**Confidence:** 5
**Clarity:** Yes, the paper is well written and ea…

**Strengths:**

1. Comprehensive Exploration: The paper addresses an important and challenging problem in computer vision—video action recognition under occlusion. It provides a comprehensive exploration of the impact of occlusion on different deep learning models, covering both transformer-based and CNN-based architectures.
2. Benchmark Datasets: The authors propose three benchmark datasets (K-400-O, UCF-101-O, and UCF-101-Y-OCC) specifically designed to study occlusion effects in video action recognition. These datasets include both synthetic and real-world occlusions, enabling researchers to conduct systematic and meaningful evaluations.
3. Transformer-Based Compositional Model: The introduction of CTx-Net, a novel compositional model based on transformers, is a noteworthy contribution. The model's demonstrated robustness to occlusion in both synthetic and natural scenarios showcases its potential for addressing the challenges posed by occluded videos.
4. Lack of Generalization Insights: The paper uncovers interesting insights into the lack of generalization among current state-of-the-art models when confronted with occlusion. This finding raises awareness about the need for improved model architectures and data augmentation techniques.

**Additional Feedback:**

Overall, I think the problem the authors are trying to solve is a very important problem and their experimental analysis shows how the current methods are lacking in that regard. I hope the authors address my concerns and that will help me to further improve my rating of the paper.

**Correctness:**

Yes, the dataset is constructed in a sound way and the evaluation methods seem appropriate.

**Documentation:**

There is sufficient detail on data collection, it is available, ethical (use of synthetic data) and the technical details along with code submission support reproducibility.

**Ethics:**

No, I do not suspect any ethical concerns.

**Limitations:**

1. Missing relevant literature: There has been prior work on occlusion handling in the skeleton action recognition field which I think needs to referenced here (for example http://openaccess.thecvf.com/content_ICCV_2019/html/Li_Making_the_Invisible_Visible_Action_Recognition_Through_Walls_and_Occlusions_ICCV_2019_paper.html, https://ieeexplore.ieee.org/document/9667032, https://ieeexplore.ieee.org/document/9162143 etc). Then there needs to be discussion of occlusion handling in the video domain in general (https://openaccess.thecvf.com/content_ICCV_2019/papers/Cheng_Occlusion-Aware_Networks_for_3D_Human_Pose_Estimation_in_Video_ICCV_2019_paper.pdf for pose estimation, https://link.springer.com/article/10.1007/s11263-022-01629-1 video instance segmentation ). Further, since there is talk about augmentation techniques I would talk about synthetic augmentation techniques such as cutting and pasting objects/humans etc (https://link.springer.com/chapter/10.1007/978-3-031-19821-2_14, https://www.sciencedirect.com/science/article/pii/S1077314222001758?casa_token=1_Lh7_gPAPgAAAAA:PtOZSRXkgZjWhdaflAtkjklAaaOqODVKCGSCug-aj1GRSNTH_KpPxyf6VLZpGsSWARLIOQY3tA etc). I think in general, the related work needs a bit of work to strengthen the paper.
2. I'm not fully convinced with the reasoning for the drop in Training on occluded data performance shown in Table 5. Can the authors strengthen their proposed hypothesis and convince me more?
3. In a similar fashion, I wonder what the authors think is the reason for the drop in performance of CTx-Net-mae.
4. There seems to be missing results of mvit v2 and videomae in Table 3 and 4.

**Opportunities For Improvement:**

1. Visualization: There are a lot of tables in the paper and some figures for evaluation such as the use of a confusion matrix which is difficult to really compare against due to the low resolution. I would suggest keeping only the last two figures in the confusion matrix and then comparing these two against each other would be all that is needed.
2. Graphs for tables: I'd suggest converting for instance Table 4 into a series of bar graphs to make it much easier to illustrate how the proposed method does much better than relevant work.

**Relation To Prior Work:**

I'm a bit concerned about the section on the prior work and have mentioned this in my limitations.

**Summary And Contributions:**

While significant progress has been made in the field of action recognition, the robustness of deep learning models, particularly in the presence of occlusion, has not been thoroughly explored. To address this, the authors propose three benchmark datasets, including synthetic and real-world occlusions, to analyze the impact of occlusion on video action recognition models. They investigate seven different models, comparing transformer-based and CNN-based architectures, and find that transformer-based models are more robust to occlusion. Additionally, they examine the effectiveness of data augmentation techniques and propose a compositional model called CTx-Net, which shows promising results in handling occlusion across both synthetic and natural scenarios.

---

> ### Author Response · Authors · 2023-08-22
>
> We thank the reviewer for their time, insightful comments, and questions. We have provided our responses below.
>
> 1.1  Clearer visualization for confusion matrix\
> We have made the visualizations for confusion maps clearer as per your suggestions, and have made the changes suggested in the updated submission.
>
> 1.2  Better visualization of results presented in Tables 3 and 4 \
> We greatly appreciate your recommendation to convert Table 4 into bar graphs to enhance clarity. While we carefully considered your suggestion, we encountered a challenge due to the relatively close performance of CTx-Net and MViT across both Tables 3 and 4. This similarity made it difficult to create bar graphs that effectively convey the performance gap in a meaningful manner. However, we've generated bar graphs for Table 3 and included them in the updated supplementary material (folder). If you believe that these bar graphs could better serve the purpose of the paper, we are open to replacing them accordingly. Your insights have been invaluable in guiding our efforts to improve the visual representation of our results.
>
> 2.1  Missing Relevant Literatures\
> Thank you for sharing these works. We have updated Section 2 to reflect these changes.
>
> 2.2 Drop in performance on training with occluded data \
> Apologies for any ambiguity in the explanation provided. We appreciate your query and aim to provide a more robust rationale for the observed drop in performance when training on occluded data, as illustrated in Table 5.CTx-Net employs a class mixture model (CMM) to gauge the likelihood of a particular feature belonging to a specific class. It is crucial to consider that CMM models the diverse variations of actions, including occluded instances. When occluded data is incorporated into the CMM learning process, the model might inadvertently associate occluded features with the same underlying motion patterns. Consequently, the CMM could learn to perceive occluded features as variations of the overall motion, which inadvertently hampers its capability to accurately classify unoccluded features. We have taken this insight into account and incorporated it into our revised submission, aiming to provide a more compelling and comprehensive understanding of the observed phenomenon.
>
> 2.3 Drop in Performance of CTx-Mae \
> We appreciate the reviewer for pointing to this interesting point about the decline in performance observed for CTx-Net-mae. To shed light on this, we performed dedicated experiments with Video MAE, and the outcomes have been incorporated into Tables 3 and 4.Upon closer inspection, it becomes evident that CTx-Net-mae exhibits notably inferior performance on specific occluders, as indicated in Table 4, when compared to MViT. The underlying reason for this decline can be attributed to the fact that CTx-Net-mae relies on a pre-trained and frozen Video MAE model. This constraint, in conjunction with the specificities of the dataset and the occlusion scenarios, contributes to the observed reduction in performance.
>
> | Model    | S1   | S2   | S3   | S4   | AA-S | M1   | M2   | M3   | M4   | AA-M |
> |----------|------|------|------|------|------|------|------|------|------|------|
> | MViTv2   | 85.8 | 83.7 | 75.2 | 82.1 | 82.6 | 83.9 | 77.3 | 81.3 | 88.3 | 82.7 |
> | VideoMAE | 79.8 | 85.2 | 73.8 | 78.2 | 79.2 | 84.1 | 67.3 | 78.1 | 89.1 | 79.6 |
>
> | Model    | L0   | L1-S | L1-D | L2-S | L2-D | L3-S | L3-D | Avg Acc |
> |----------|------|------|------|------|------|------|------|---------|
> | MViTv2   | 96.5 | 94.5 | 87.2 | 89.5 | 82.3 | 81.3 | 72.4 | 86.2    |
> | VideoMAE | 96.0 | 94.7 | 87.8 | 91.2 | 80.1 | 81.5 | 66.7 | 85.4    |
>
>
> 2.4 Missing results for MViTv2 and VideoMAE in tables 3 and 4\
> The results for MVitv2 and Video MAE have been added to Tables 3 and 4.
>
> 3.1 Relation to Prior Work
> We have updated section 2 to address this concern

---

> > ### Comment · Reviewer_pkpL · 2023-08-24
> > **Responding to rebuttal**
> >
> > I'm happy with the response provided. I would still say a deeper analysis of responses in 2.2 and 2.3 above would be helpful for the paper. By deeper, I mean having visualizations to back up the hypothesis (can be activation maps or examples of failures etc). This could be added in the supplementary and certainly make the paper much stronger.

---

> > > ### Author Response · Authors · 2023-08-25
> > >
> > > We thank the reviewer for their valuable time and effort in reading our response. We are glad that we were able to address most of the queries. Regarding 2.1 and 2.2, we agree with the reviewer that a deeper analysis will strengthen the work. Based on reviewers suggestions, we have now added more analysis with visualisations and examples of failures in the supplementary.
> > >
> > > 2.1: We have added occlusion map visualisation for CTx-Net + data aug for unoccluded data. From figure 10 in the supplementary material, we can observe that the CTx-Net (data aug) model is identifying certain features as occluders even though they are unoccluded. This strengthens our hypothesis that training CMM on occluded data would result in lower accuracy in detecting class relevant features.
> > >
> > > 2.2: We have also included videos on which Video MAE fails although MViT gives the correct predictions. This also strengthens the results in Table 3 and 4 (main paper) where Video MAE provides poorer performance as compared to MViT.
> > >
> > > We hope this analysis addresses the concerns raised by the reviewer. We will be more than happy to discuss this further. Thank you for your valuable suggestions.

---

### Official Review · Reviewer_6zfo · 2023-07-30
**Benchmarking 7 action recognition models on proposed 2 synthetic + 1 real-world occlusion datasets**

**Rating:** 7
**Confidence:** 3
**Clarity:** Needs explanations for experimental s…

**Strengths:**

- State of the art models are used for comparison.
- Three new datasets are released.
- This paper addresses an open research gap on occlusion based action recognition.
- It shows how compositionally based models perform better than their counterparts.
- It also contributes in terms of a method for occlusion based action recognition.

**Additional Feedback:**

- Are there any transfer learning results from the synthetic to the real-world dataset?
- Role of occlusion based augmentations in the contrastive learning approach is mostly unexplored. It may be discussed based on the available results.
- In Table 1,2, what datasets are used for pretraining CTxNet? It may be discussed as how pretraining on Kinetics400 vs UCF101 affects and which is more useful.

**Correctness:**

- In Table 1, please explain: R2P1D and MViT robustness with augmentation is better than without augmentation for UCF-101-O. However, its inverse is true for UCF-101Y-OCC?
- From the representative videos and images, UCF-101Y-OCC looks like a less challenging dataset than UCF-101-O. Then please explain why is that the performance of models on synthetic occlusion datasets degrades less in comparison to the real-world dataset.

**Documentation:**

Sufficient details are there except for pretraining protocol and transfer learning.

**Opportunities For Improvement:**

- It is ambiguous to name the real-world dataset UCF101 since 101 implies the number of classes, which in this case is 19. Therefore, it should be renamed to anything (like Y19OCC, UCF19-Y-OCC, etc.) but UCF101.
- Please indicate which models are trained fully supervised and which are partially supervised. It is a bit confusing at present.
- The proposed CTxNet baseline is similar to [1] except that it uses a transformer than CNN; please discuss the difference between both.
- At present, there is a lot of ambiguity regarding the protocol for pretraining, datasets used for pretraining, transfer learning data proportions, and whether supervised or self-supervised pretraining. Pretraining details should be given for all uses. It is not clear which results correspond to pretrained and which don't. Also, it is not clear which datasets were used to pretrain a particular model and what was the protocol for finetuning. All these details need to be mentioned for each experiment and each model used.
- Some uniformity is missing in results, such as augmentation results are not present for more interesting methods like MViTv2, VideoMAE, and the proposed CTx-Net in Table 1 and also in Table 2; MViTv2 and VideoMAE are absent in Table 3; Kinetics400 dataset and other models are absent in Table 5 ablations; and CTx-Net in Figure 6.

[1] Kortylewski, A., He, J., Liu, Q., & Yuille, A. L. (2020). Compositional convolutional neural networks: A deep architecture with innate robustness to partial occlusion. In Proceedings of the IEEE/CVF Conference on Computer Vision and Pattern Recognition (pp. 8940-8949).

**Relation To Prior Work:**

- In literature, please discuss the previous works on occlusion based action recognition and related datasets.

**Summary And Contributions:**

This paper benchmarks seven action recognition models on three occlusion datasets. It attempts to fill the research gap on benchmarking of action recognition models under occlusion. Of the three datasets, two are synthetic that were made by putting objects from Pascal-VOC dataset in 0-60% area of videos in UCF101 and Kintetics400 datasets. The natural occlusion dataset was made by selecting videos from Youtube in 19 classes (30 samples per class). Authors also proposed a transformer based compositional model (CTx-Net), which is claimed to disentangle occluders from the action. All models were evaluated to test their resilience to different kinds of occlusions, effect of pretraining, and effect of transformers over CNNs. SOTA methods were used for comparison. Overall, the work addresses the research gap on occlusion based action recognition and shows how compositionality improves performance. This paper provides new benchmarks and valuable conclusions to the problem of robust action recognition.

---

> ### Author Response · Authors · 2023-08-22
>
> We thank the reviewer for their time, insightful comments, and questions. We have provided our responses below.
>
> 1.1  Ambiguous dataset name\
> Thank you for this suggestion. We have updated the name of the dataset to UCF-19-Y-OCC (UCF-101-Y-OCC to UCF-19-Y-OCC) to avoid this ambiguity.
>
> 1.2  Model Pretraining\
> We sincerely appreciate the reviewer's concern regarding the training procedures of the studied models. Except for VideoMAE, all the studied models use fully  supervised training on the training split of K-400 dataset. To obtain results on UCF-101, we initialized all models with weights acquired from the K-400 training phase. Subsequently, fine-tuning was carried out in a fully supervised manner on the training set of the UCF-101 dataset. VideoM.AE was subject to additional self-supervised training on K-400 followed by supervised training on the training split of K-400. For UCF-101, weights garnered from the fully supervised training on the K-400 training set are used for the fine-tuning process. In the case of augmented models, initialization occurred with pre-trained K-400 weights. After this, synthetic occlusion is used on video samples as augmentation for fine tuning the model on  UCF-101 training split. More details about augmentations are provided in section 5 of the submission.  We acknowledge that the clarity of this information could benefit from improvement and have addressed it accordingly in the updated submission.
>
> 1.3 Comparison with [1]\
> In [1], the concept of compositional networks for occlusion was introduced, with a primary focus on image classification. Our work has extended this framework into the video domain, specifically targeting spatio-temporal action detection. Noteworthy distinctions between the two include:
> * Choice of Backbone: Unlike [1] we utilize a transformer in our CTx-Net which allows us to effectively capture spatio-temporal features inherent to video data.
> * Occluder Kernel Types: While [1] exclusively employed spatial occluder kernels, we propose a more comprehensive approach. Our methodology encompasses both spatial and spatio-temporal occluder kernels, enabling the incorporation of temporal characteristics during the occlusion modeling process.
> * Feature Integration: We incorporate a combination of visual features derived from the transformer token sequence alongside class tokens, which is facilitated by the transformers architecture.
>
> 1.4 Pretraining Protocol\
> Thanks for raising this point. As described in response to the second query, we have now updated the pretraining details. As mentioned, pretrained weights from the K-400 dataset are used for initialization of model weights for fine-tuning. Fine-tuning is done in a fully supervised manner on the target dataset.
>
> 1.5 Missing Results \
> We greatly appreciate the reviewer's observation regarding the absence of uniformity in the presentation of results. To address this concern, we have made the following enhancements\
> * We have incorporated augmented results for Video MAE and MViTv2 into Table 1, as suggested.
> * As per your suggestion, we have included the outcomes for MViTv2 and VideoMAE in both Tables 3 and  4.
>
> | Model         | UCF-101 (Top 1 Acc) | UCF-101-O (Top 1 Acc) | UCF-101-O($ \gamma^a $) | UCF-101-O ($ \gamma^r $)  | UCF-19-Y-OCC (Top-1-Acc) | UCF-19-Y-OCC ($ \gamma^a $) | UCF-19-Y-OCC ($ \gamma^r $) |
> |---------------|---------------------|-----------------------|-------------------------|---------------------------|--------------------------|-----------------------------|-----------------------------|
> | MViTv2(aug)   | 95.7                | 88.3                  | 0.92                    | 0.92                      | 65.3                     | 0.69                        | 0.69                        |
> | VideoMAE(aug) | 95.8                | 87.1                  | 0.91                    | 0.91                      | 64.2                     | 0.68                        | 0.67                        |
>
> | Model    | L0   | L1-S | L1-D | L2-S | L2-D | L3-S | L3-D | Avg Acc |
> |----------|------|------|------|------|------|------|------|---------|
> | MViTv2   | 96.5 | 94.5 | 87.2 | 89.5 | 82.3 | 81.3 | 72.4 | 86.2    |
> | VideoMAE | 96.0 | 94.7 | 87.8 | 91.2 | 80.1 | 81.5 | 66.7 | 85.4    |
>
> | Model    | L0   | L1-S | L1-D | L2-S | L2-D | L3-S | L3-D | Avg Acc |
> |----------|------|------|------|------|------|------|------|---------|
> | MViTv2   | 96.5 | 94.5 | 87.2 | 89.5 | 82.3 | 81.3 | 72.4 | 86.2    |
> | VideoMAE | 96.0 | 94.7 | 87.8 | 91.2 | 80.1 | 81.5 | 66.7 | 85.4    |
>
>
> Due to the extended training duration associated with K-400, we are actively working to include augmented MViT results for Table 2,

---

> > ### Author Response · Authors · 2023-08-22
> >
> > 2,1 Robustness on UCF-101-O and UCF-19-Y-OCC\
> > The noted observation is quite interesting. In the context of UCF-101-O, which is a dataset subject to synthetic augmentation, the presence of synthetic occluders induces a distribution shift that is relatively consistent across training and test set. As a result, the models' robustness improves in this context.However, the scenario differs with UCF-101Y-OCC. Despite employing synthetic occluders in this study, the distribution shift they induce proves inadequate for ensuring robustness against real-world occlusion challenges. This leads to the conclusion that augmented models exhibit lesser robustness compared to their original counterparts in such scenarios.Furthermore, this observation underscores the limitation of simplistic augmentation-based strategies in attaining substantial real-world occlusion robustness. It becomes evident that more sophisticated approaches are imperative to achieve the desired level of robustness against occlusion in practical settings.
> >
> > 2.2   Comparison of performance of models on UCF-19-Y-OCC and UCf-101-O\
> > Indeed, your observation regarding the apparent contrast in the challenge level between UCF-101Y-OCC and UCF-101-O is noteworthy. This paper's key insight is precisely aligned with this observation: the distribution shift generated by synthetic occlusion falls short in effectively addressing the diverse array of real-world occlusion scenarios. Tables 1, 2, 3, and 4 provide substantial evidence to support this conclusion. These tables reveal that the motion and shape characteristics of occluders play a pivotal role, transcending mere occlusion severity. Notably, the occluders encountered in real-world datasets exhibit a significantly broader range of shapes and motions, which contributes to the higher level of complexity in handling real-world occlusions.This underscores the inherent limitations of relying solely on synthetic occlusion to prepare models for real-world occlusion challenges, underlining the need to consider diverse factors beyond mere severity.
> >
> > 3.1   Relation to prior work
> > We have added previous contributions related to occlusion in the video domain and the image domain in Section 2.1 to address this point.
> >
> > 4.1 Transfer Learning Results \
> > The results presented in Table 1 showcase the performance of models trained on augmented data using synthetic occlusion. Specifically, the results highlighted under UCF-19-OCC demonstrate the transfer learning outcomes achieved when applying these models to the real-world occlusion dataset.
> >
> > 4.2 The role of occlusion in contrastive learning approaches\
> > Contrastive learning and the role of occlusion are exciting directions for future work. Our work consists of models trained in a fully supervised manner, apart from Video MAE. So we are unable to draw conclusions regarding contrastive learning settings.
> >
> > 4.3 CTx-Net Pretraining \
> > The backbone utilized for CTx-Net in Table 1 involves the employment of MViT that has undergone training on Kinetics400 (K-400), followed by fine-tuning on UCF-101. It's noteworthy that maintaining consistent class distributions between the backbone and the class mixture model is essential for models like Class mixture models, as class encoding is integrated with the features. Moreover, to shed light on the comparative impact of different training datasets, we have introduced results for CTx-Net where the backbone is exclusively trained on UCF-101. These results are presented in Figure 6 , providing insights into the influence of pretraining on Kinetics400 versus UCF-101 on the performance of CTx-Net.

---

> > > ### Comment · Reviewer_6zfo · 2023-08-23
> > >
> > > Most of my queries have been addressed. The missing results have been incorporated. My rating stays the same for the following reason:
> > > - I am still not convinced with the response 2.2. As per my observation, synthetic occlusions also have motion and shape deformations which appear more challenging than the natural dataset.

---

> > > > ### Author Response · Authors · 2023-08-23
> > > >
> > > > We thank the reviewer for their valuable time and effort in reading our response. We are glad that we were able to address most of the queries. We apologies for non-convincing response for performance gap between synthetic and realistic evaluations.
> > > >
> > > > We agree with the reviewer that synthetic occlusions also have motion and shape deformations which can be challenging, but still it can be hard to mimic the real-world occlusions. Therefore, we also included the realistic occlusion dataset. The reviewers observation has also been found in some other recent studies, such as [R1], where the performance on real-world dataset for distribution shift was lower in comparison with synthetic distribution shifts. This lower performance can also be attributed to distribution shift between original UCF-101 dataset and proposed occluded dataset due to different time-frame when they were curated. However, since the performance trend across all models is consistent with synthetic occlusions, it does not affect the findings in this study. We believe the proposed realistic occlusion dataset will be very beneficial to study and benchmark action recognition under occlusion in videos.
> > > >
> > > > We hope this clarifies the doubt raised by the reviewer. We will be more than happy to discuss this further. Thank you for your valuable insights.
> > > >
> > > > [R1] Schiappa, Madeline Chantry, et al. "A Large-Scale Robustness Analysis of Video Action Recognition Models." Proceedings of the IEEE/CVF Conference on Computer Vision and Pattern Recognition. 2023.

---

> > > > > ### Comment · Reviewer_6zfo · 2023-08-24
> > > > >
> > > > > Thank you for addressing my doubts. I am satisfied with the response.

---

### Author Response · Authors · 2023-08-22

We thank all the reviewers for their precious time and insightful comments. We appreciate that the reviewers recognize the strengths of our work in terms of benchmarking, the contribution of synthetic and naturally occluded datasets, and the intuitive insights the analysis provides. We also acknowledge the valid concerns of reviewers regarding missing literature in recent works sections and a lack of pretraining details.

Here is a summary of how we have addressed major points raised by the reviews.
*  Adding results for models VideoMAE and MViTv2 for Tables 3 and 4
*  We add results of augmented VideoMAE and MViTv2 for Table 1
*  We provide detailed pre-training details
*  We added more relevant works to the recent works section
*   Add visualizations for the datasets
*   Add the data collection procedure of real world dataset

We hope this should be able to address the reviewers' concerns, and we are happy to engage in further discussion to resolve any more concerns that the reviewers might have.

---

### Decision · Program_Chairs · 2023-09-22

**Decision:**

Accept (Poster)

**Comment:**

The paper received reviews from 4 different reviewers and most of them are positive. During rebuttal, the author(s) provide additional experiments and clarifications. Most concerns are addressed and the negative reviewer raises their score to positive. The meta reviewer reads reviews and discussions and recommend to accept the paper. The meta reviewer recommends the author(s) incorporate comments from the reviewers to strengthen the paper in camera-ready version.